# MILP Formulation for Solving and Initializing MINLP Problems Applied to Retrofit and Synthesis of Hydrogen Networks

**Patrícia R. da Silva [1,\*], Marcelo E. Aragão [2], Jorge O. Trierweiler [1] and Luciane F. Trierweiler [1]**

[1] Group of Intensification, Modeling, Simulation, Control, and Optimization of Processes, Chemical Engineering Department, Federal University of Rio Grande do Sul (UFRGS), R. Eng. Luiz Englert, s/n, Campus Central, Porto Alegre 90040-060, RS, Brazil; jorge@enq.ufrgs.br (J.O.T.); luciane@enq.ufrgs.br (L.F.T.)

[2] Food and Chemistry School, Federal University of Rio Grande (FURG), R. Barão do Cahy, 125, Cidade Alta, Santo Antônio da Patrulha 95500-000, RS, Brazil; escobar@furg.br

\* Correspondence: patriciars@enq.ufrgs.br

**Abstract:** The demand for hydrogen in refineries is growing due to its importance as a sulfur capture element. Therefore, hydrogen management is critical for fulfilling demands as efficiently as possible. Through mathematical modeling, hydrogen network management can be better performed. Cost-efficient Mixed-Integer Linear Programming (MILP) and Mixed-Integer Nonlinear Programming (MINLP) optimization models for (re)designing were proposed and implemented in GAMS with two case studies. Linear programming has the limitation of no stream mixing allowed; therefore, to overcome this limitation, an algorithm-based procedure called the Virtual Compressor Approach was proposed. Based on the MILP optimal solution obtained, the streams and compressors were merged. As a result, the number of compressors was reduced, along with the inherent investment costs. An operational cost reduction of more than 28% (example 1) and 26% (example 2) was obtained with a linear model. The optimal MILP solution after rearranging compressors was then provided as a good starting point to the MINLP. The operating costs were decreased by more than 31% (example 1) and 32% (example 2). Most of the cost reduction was obtained only with the usage of the MILP model. Besides, a higher level of cost reduction was only obtained when the linear model was used as the starting point.

**Keywords:** hydrogen network; mathematical programming; initialization strategy; MILP optimization; MINLP optimization; virtual compressor approach

## 1. Introduction

Hydrogen has a prominent role in the refining industry, as both its production and its recovery are essential steps. Hydrogen consumption in oil refining increased from approximately 7 million tons in 1980 to 38 million tons in 2018 [1]. Its importance is sustained by three factors: (i) the increase in the processing of heavier oils with high levels of sulfur and nitrogen; (ii) the increase in environmental constraints; and (iii) the production of derivatives of higher added value [2,3]. Due to this trend, it is necessary to use more efficient hydrogen within the petroleum refining process.

A hydrogen network consists of hydrogen-producing units, hydrogen-consuming units, and purification units, capable of purifying hydrogen to achieve the required purity. The hydrogen generation units (HGU) have become increasingly present in refineries due the importance of hydrotreatment units (HDT) because its function is to supply the hydrogen demand complementing those generated in the catalytic reform. The steam reform is the primary process used at the industrial scale to obtain hydrogen as a primary product. Catalytic reform and purge gas can be used as a

secondary source of hydrogen. The main hydrogen-consuming units are hydrotreating, which uses hydrogen to improve the quality of naphtha, kerosene, solvents in general, diesel oil, heavy gas oils, paraffin, and lubricating oils [4]. The management of the hydrogen network in a refinery implies in the material balance at all these units.

The need for optimization of the hydrogen network in refineries was recognized in the 1990s, because, usually, the amount of hydrogen produced is higher than the amount consumed. This excess is usually incorporated into the fuel gas system or burned directly into the flare. Therefore, it is necessary to have greater control in the sources and consumers of hydrogen through network management as a whole, because it is not economically feasible to produce and burn the product with an excellent added value [5]. It is known that the cost of hydrogen is the second-highest cost in a refinery, behind only the cost of crude oil [6]. Therefore, savings in terms of the amount of hydrogen consumed and the operating cost of the network have great economic appeal.

Since then, many methodologies have emerged to accomplish it. In general, these methodologies can be divided into two categories: pinch methods and optimization methods (deterministic in this case) as mathematical programming approaches based on network design [7]. Graphical methods provide an essential insight into the integration of the refinery process and provide theoretical goals for minimum hydrogen use. As oil refining and the hydrogen network involve many restrictions, they must be considered during network modeling and optimization, such as pressure, impurities, and equipment capacity. However, in graphic methods, this is not possible, as only flow and purity restrictions are considered. Therefore, mathematical programming is the best alternative and the most used, providing more realistic results and networks [8].

In this work, a mathematical programming approach was used to develop a model to solve the problem of hydrogen network optimization based on operating costs and constraints. The idea is to apply the proposed model to existing networks. The optimization allows the possibility of including new equipment and finding better ways of connection between units. For linear optimization, a compressor rearrangement technique was proposed in this work to decrease the capital cost. It is called Virtual Compressor Approach (VCA). The methodology was proposed to make the linear model competitive and satisfactory for the retrofit of hydrogen networks, due to its advantages and characteristics. Besides, a nonlinear model was also developed for comparison, with an initialization strategy using the MILP solution. This proposal was developed to facilitate the resolution of nonlinear and obtaining more competitive hydrogen networks.

## 2. Literature Review

Previous works on hydrogen distribution management and analysis using a linear programming model, based on the graphical analysis of the pinch method, were found in the literature. Towler et al [9] proposed a linear model to optimize a hydrogen network, aiming to minimize the total hydrogen import as an external utility. Two procedures for problem relaxation were proposed. The disadvantages of this method are that pressure constraints are negligible, and the flow merging must be performed manually [9]. Fonseca et al. [10] employed the linear programming model to optimize the hydrogen network of a refinery taking account pressure considerations and achieved a 30% reduction in utility use with the objective function minimizing the total flow rate of fresh hydrogen from a hydrogen plant [10].

Considering nonlinear programming (NLP), Hallale and Liu [11], in addition to mentioning the graphical pinch method, developed a nonlinear mathematical model to reduce the hydrogen consumption of the network. The model took into account pressure constraints, existing compressors, and a strategy to install a purifier. The objective function was to minimize the total cost, including operating and capital costs [11]. Liu and Zhang [12] developed a systematic procedure for integrating purification in hydrogen network design. For this, an MINLP (Mixed-Integer Nonlinear Programming) model for purifier selection and integration was used, and with linear relaxation of bilinear forms MINLP model was solved first as MILP because of the advantages of using linear models for problem

solution [12]. Kumar et al. [13] developed mathematical models (LP (linear programming), NLP, MILP (Mixed-Integer Linear Programming), and MINLP) to obtain the best optimization problem in two case studies. Comparing MINLP and NLP for case 1, MINLP showed a more significant reduction in operating costs and equal capital costs. For case 2, the formulations LP, NLP, and MILP were compared. The NLP model imports less hydrogen and features a more realistic network than the others. The conclusions were that mixed-integer linear and nonlinear programming models are considerably better than linear because it provides the less complicated and more realistic refinery system, and MINLP can include complexities as compressors, purity constraints, and pressure constraints [13].

Liao et al. [14] developed an MINLP model using an existing hydrogen network with a purifier. The objective function was the total annual cost, and the model was solved in GAMS (General Algebraic Modeling System) using DICOPT. The MINLP problem is decomposed into a series of NLP and MILP solvers. The total annual cost decrease by 22.6% and both the new compressor and PSA were incorporated [14]. Birjandi et al. [15] developed a methodology for the optimization of a hydrogen network based on a simultaneously resolved MINLP and NLP problem. Linearization techniques for nonlinear models were used to facilitate resolution by transforming nonlinear equality constraints into inequality constraints. Global optimization has reduced operating costs [15]. Matijasevic [16] presented a hydrogen network integration methodology for a case study of a local refinery. The minimum consumption of hydrogen was determined by pinch analysis. Then, the superstructure was modeled using a nonlinear mathematical model whose objective function was to minimize total operating costs. The problem was solved with the GAMS software [16].

Unlike what was found in the literature, this paper developed a cost-efficient MILP and MINLP optimization models for (re)designing of hydrogen networks or a new project. The main difference from the MILP model to the MINLP is that it is not possible to mix streams in the compressors as it generates nonlinearity. To reduce the cost of capital from the MILP, in this work, a compressor-retrofitting tool was proposed respecting the nominal capacities. Also, to facilitate the resolution of the nonlinear formulation, an initialization strategy was used using the linear solution as a feasible starting point.

## 3. Mathematical Programming Approaches

Mathematical programming based on superstructure has advantages over pinch, in that it considers numerous limitations and variables when looking for solutions to the optimization problem. Limitations such as pressure, capacity, purity, operating costs, and investments in new equipment are some of the restrictions that may be included in the mathematical model formulation. The methodology to develop mathematical programming would be the development of the superstructure, including the sources, consumers, existing compressors, and purifiers. The formulation of the mathematical model also includes the objective function to be minimized or maximized subject to the set of constraints, the initialization strategy, and the resolution of the optimization problem. Typically, the objective function is the total annual cost of the hydrogen network [8].

Generally, the optimization problem can be formulated as a linear programming, mixed linear programming, nonlinear programming, or mixed-integer nonlinear programming problem. If linear combinations of variables can express the objective function and constraints, it is a linear optimization problem. Otherwise, the optimization problem is nonlinear. There are many optimization software used to solve optimization problems and already include algorithms called solvers [17].

Network management through mathematical modeling can be applied to an existing fixed topology, or to develop a new hydrogen network design. Thus, the approach of this article is based on the evaluation of the model developed for initial hydrogen network projects, through the validation with networks presented in articles already published. New equipment is considered, and the problem then becomes MILP or MINLP. Although the focus is operational, the problem addressed here is broader and has a significant industrial interest. The primary purpose of managing hydrogen networks is their production with minimum slack. Excess hydrogen production must be minimized, first because

hydrogen is not easy to handle or store, and second, because it is not economically viable since the excess must be burned as fuel and furnaces and other processes.

The MINLP problems are more challenging to solve because they combine the NLP and MILP models and their characteristics. However, they result in more realistic networks and include several additional restrictions. According to the literature review, the use of MILP is not very recurrent, although when used, it presents significant results. Most articles found in the literature use nonlinear models for hydrogen network optimization. The advantages of using MILP is the linearity that facilitates the resolution of the optimization problem and the modeling of the logical constraints made in this article, which were not found in the literature. MILP problems are easier to converge to a global solution, since all the subproblems, for fixed binaries, are linear solved to global optimality [18,19].

### 3.1. Problem Statement

The problem to be addressed in this paper can be stated as follows: (i) a set of sources $i \in$ hydrogen sources (*HS*), (ii) a set of consumers j $\in$ hydrogen consumers (*HC*), and (iii) a set of purifiers $k \in$ hydrogen purifiers ($HP = OHP \cup NHP$), considering the existing purifiers, *OHU*, and the new purifiers, *NHP*. In the case of nonlinear formulation, there is still a set of compressors c $\in$ hydrogen compressors ($HCP = OHCP \cup NHCP,$) considering the existing compressors *OHCP* and new compressors *NHCP*. Figure 1 shows the two superstructures considered in this problem for the linear formulation (Figure 1a) and the nonlinear formulation (Figure 1b).

For each source, the maximum and minimum flowrate, as well as the hydrogen composition, and the outlet pressure are given. For each consumer, the inlet flowrate demand, pressure, and composition, the outlet purge flow, pressure, and composition are given. For each purifier, the maximum flow capacity, the composition of purified flowrate and purge flowrate, the pressure of purification, and the hydrogen recovery are given. It is also considered a fuel system in which waste streams can be burned and used as fuel to the process. For the existing networks, also given are the existing lines (unit connections), the distance between the units if informed, and the existing compressors (capacity and pressures) and purifiers.

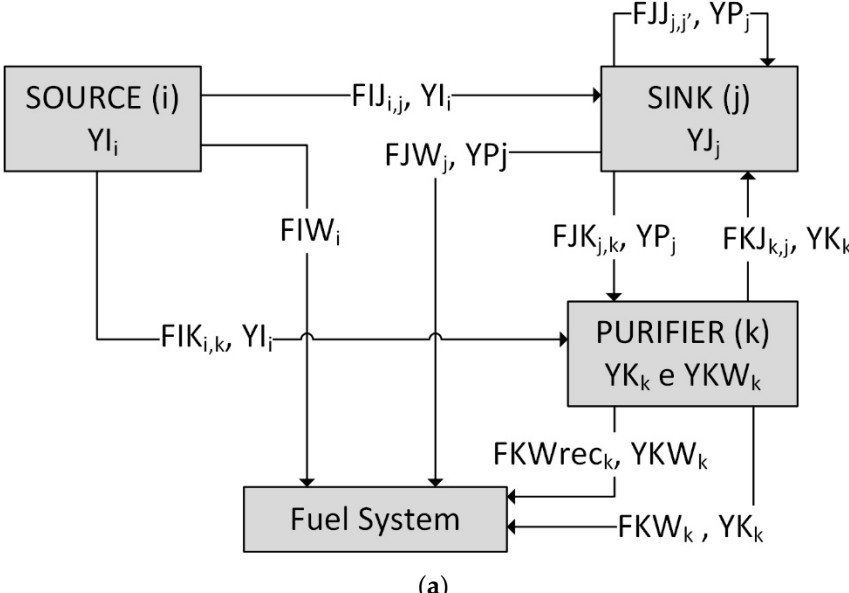

(**a**)

**Figure 1.** *Cont.*

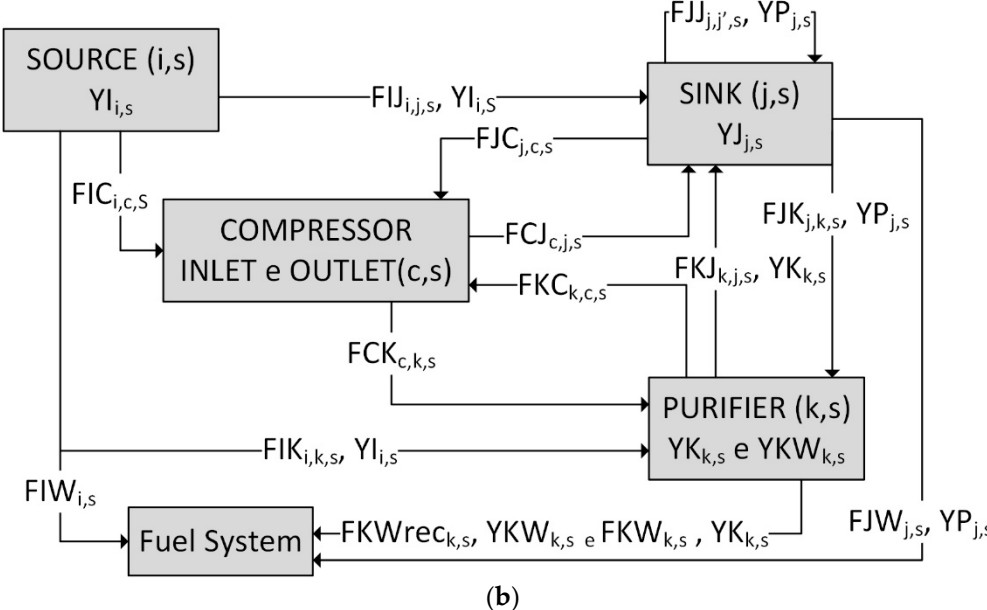

(**b**)

**Figure 1.** (**a**) Scheme of the Superstructure developed for the Mixed-Integer Linear Programming (MILP) problem. (**b**) Scheme of the Superstructure developed for the Mixed-Integer Nonlinear Programming (MINLP) problem.

The optimization problem is subject to the material balances and process operating constraints. For the retrofit case, process modifications are allowed to reduce the total operating costs (the objective function), despite the investment costs due to the installation of new pipelines, compressors, and possibly new purifiers.

### 3.2. Mathematical Model: MILP Formulation

Figure 1a shows the superstructure and all the possible connections among these four units between sources and consumers, sources and purifiers (existing and new ones), as well as flows between consumers and the purifying units for sources $i$ and consumers $j$. The first step for the modeling development is to define which units are involved in the hydrogen network, for instance, which units provide hydrogen, which units consume hydrogen and the existing purifiers, and the potential purifiers that should be considered in the model.

The optimization problem of hydrogen network design in this work can be summarized as follows: the superstructure is formed by a set of sources of hydrogen $i$, a set of hydrogen consumers $j$ and set of units of hydrogen purification $k$, account for the existing and new purifiers. The hydrogen sources have their minimum and maximum flow according to their capacity ($FH2I_{i,min}$ e $FH2I_{i,max}$) as well as their hydrogen purity ($YI_i$). The hydrogen-rich stream can be sent to the consumers $j$ ($FIJ_{i,j}$), to purification units $k$ ($FIK_{i,k}$), or can be sent to the fuel system ($FIW_i$). The consumer's units also have their known, and constant input required flows for the process ($FJ_j$), as well as its hydrogen purity ($YJ_j$), in addition to the outflows ($FP_j$) and hydrogen purity ($YP_j$), according to the hydrogen consumption of each specific process. The outlet flows from the consumers can be sent to purification ($FJK_{j,k}$), can be used as a source for other consumers ($FJJ_{j,j'}$) or can be sent to the fuel system ($FJW_j$) to be used as the burning fuel. The purifying units have a known hydrogen recovery ratio ($rec_k$), as well as the maximum inlet flow capacity ($FPur_{max,k}$) and the constant purities of the hydrogen product pure streams ($YK_k$) and the composition for the stream of hydrogen not recovered stream ($YKW_k$). The purified hydrogen stream from the purification can be used as a source for the consumers ($FKJ_{k,j}$) who need higher purity or can be referred to the fuel system ($FKW_k$), if there is excess. The stream with the unrecovered hydrogen, $FKWrec_k$, has a small hydrogen composition, and it is sent directly to the fuel system. In this work, some considerations were made to simplify the model. The flowrates are

considered only a binary mixture of hydrogen and methane. The partial pressure of the hydrogen and the flowrate are constant at the entrance and exit of the consuming units.

### 3.2.1. Sources

The overall material balance for each source is represented by Equation (1):

$$FH2I_i = \left( \sum_{j \in HC} FIJ_{i,j} + \sum_{k \in HP} FIK_{i,k} + FIW_i \right) \forall i \in HS \tag{1}$$

where $FH2I_i$ is the total flow from each source $i$, $FIJ_{i,j}$ is the hydrogen flow from the source $i$ to the consumer $j$, $FIK_{i,k}$ is the flow from the source $i$ for the purification unit $k$, and $FIW$ is the flow from source $i$ sent to the fuel system. The available flow rate is limited by the capacity of the hydrogen generating units according to the following inequality constraints:

$$FH2I_{i,min} \leq FH2I_i \leq FH2I_{i,max} \ \forall i \in HS \tag{2}$$

### 3.2.2. Consumers

Equation (3) represents the overall material balance in the inlet of consumer units.

$$FJ_j = \sum_{i \in HS} FIJ_{i,j} + \sum_{k \in HP} FKJ_{k,j} + \sum_{j \in HC} FJJ_{j,j'} \ \forall j \in HC \tag{3}$$

where $FJ_j$ is the total flow directed to consumers, $FJJ_{j,j'}$ is the flow from one consumer $j$ to another consumer $j'$ and $FKJ_{k,j}$ is a flow rate from the purification unit $k$ for the consumer units $j$. The index $j'$ which is used for cases where there is a connection between consumers. In this case, as it is not allowed between the same unit, $j'$ must be different from $j$. The hydrogen balance is then defined by Equation (4):

$$FJ_j * YJ_j = \sum_{i \in HS} FIJ_{i,j} \times YI_i + \sum_{k \in HP} FKJ_{k,j} \times YK_k + \sum_{j \in HC} FJJ_{j,j'} \times YP_j \ \forall j \in HC \tag{4}$$

where $YJ_j$, $YI_i$, $YK_k$ and $YP_j$ are the volumetric fractions of hydrogen in the respective streams, consumer $j$, sources $i$, purifiers $k$, and purge of the consumer unit $j$. Besides, it is possible to calculate how much each consumer unit used hydrogen depending on the chemical process involved.

Equation (5) represents the overall material balance in the outlet of consumer units:

$$FP_j = FJW_j + \sum_{k \in HP} FJK_{j,k} + \sum_{j \in HC} FJJ_{j,j'} \ \forall j \in HC \tag{5}$$

where $FP_j$ is the total flow out of consumers, $FJK_{j,k}$ is the flow rate from the consumer unit $j$ for the purification unit $k$, and $FJW_j$ is the surplus flow of consumers directed to the fuel system.

### 3.2.3. Purification Units

The purification unit is used, so that process streams are purified, providing hydrogen in a given purity, such as 99.99% in the case of PSA units. The overall material balance in these units is expressed as:

$$\sum_{j \in HC} FJK_{j,k} + \sum_{i \in HS} FIK_{i,k} = \sum_{j \in HC} FKJ_{k,j} + FKW_k + FKW_{rec,k} \ \forall k \in HP \tag{6}$$

where $FKW_k$ the flow rate of the purifying unit $k$ stream rich in hydrogen routed to burning and $FKW_{rec,k}$ is the hydrogen flowrate not recovered by the purifying unit $k$ sent to the burner. The hydrogen balance for each purifier is described as follows:

$$\sum_{j\in HP} FJK_{j,k} \times YP_j + \sum_{i\in HS} FIK_{i,k} \times YI_i = \sum_{j\in HP} FKJ_{k,j} \times YK_k + FKW_k \times YK_k$$
$$+ FKW_{rec,k} \times YKW_k \; \forall k \in HP \tag{7}$$

where $YKW$ is the fraction of hydrogen in the purge stream of purified $k$. The total flow entering the purifier is limited by the capacity of the purifying unit.

$$\sum_{j\in HP} FJK_{j,k} + \sum_{i\in HS} FIK_{i,k} \leq \sum_{k} FPur_{max,k} \; \forall k \in HP \tag{8}$$

Given the hydrogen recovery of the purification unit, it is possible to calculate how much hydrogen is sent to the purge stream, i.e., the hydrogen not recovered.

$$\left( \sum_{i\in HS} FIK_{i,k} \times YI_i + \sum_{j\in HP} FJK_{j,k} \times YP_j \right) \times (1 - rec_k) = FKW_{rec} \times YKW_k \; \forall k \in HP \tag{9}$$

The total flow through the PSA ($FK_k$) can then be defined as:

$$\sum_{j\in HP} FJK_{j,k} + \sum_{i\in HS} FIK_{i,k} = FK_k \; \forall k \in HP \tag{10}$$

### 3.2.4. Logical Constraints

To consider the capital cost associated with new equipment, it is necessary to use constraint modeling, through logical propositions and disjunctions, so binary variables and logical inequality equations were included in the model with binary parameters. First, through the modeling of disjunctions, a binary variable z is associated with the existence of a particular flow $F$ (e.g., $FIJ_{i,j}$, $FKJ_{k,j}$, $FJK_{j,k}$, etc.). If the positive flowrate is greater than or equal to a small value $\varepsilon$, e.g., $\varepsilon = 10^{-5}$, the corresponding binary variable z assumes the value of 1. On the other hand, if the flowrate is lower than $\varepsilon$, the binary variable assumes the value of 0. $F_{max}$ are the flowrates between the units involved. These conditions are ensured by the following constraints:

$$\begin{cases} F \geq \varepsilon \times z \\ F \leq (\min(F_{max})) \times z \end{cases} \tag{11}$$

A binary variable $z_c$ is associated with the installation of a compressor for the corresponding flow. For this case three events must hold simultaneously: (i) there is a non-zero flow, i.e., $z = 1$; (ii) there is no compressor previously installed identified by a binary parameter $u_c$ (1 if there is an existing compressor, 0 otherwise); and (iii) there is a pressure difference between the current unit and destination unit that requires a compressor identified by a binary parameter $u_{deltaP}$ (1 if the current pressure unit is lower than the destination pressure unit, 0 otherwise).

$$z_c \geq z + u_{deltaP} + (1 - u_c) - 2 \tag{12}$$

If any of these three events is false, then there is no need to install a compressor ($z_c = 0$), which is ensured by the set of constraints described in the set of Equation (13).

$$\begin{cases} z \geq z_c \\ 1 - u_c \geq z_c \\ u_{deltaP} \geq z_c \end{cases} \tag{13}$$

A similar procedure was used to consider the investment cost of piping. A binary variable $z_h$ is associated to the need of installing a new pipeline if two events hold: (i) there exists a non-zero flow in that connection, i.e., $z = 1$; (ii) there is no pipeline previously installed identified by a binary parameter $u_h$ (1 if there is a line, 0 otherwise).

$$z_h \geq z + (1 - u_h) - 1 \tag{14}$$

If any of these two events do not hold, it must be ensured that no pipeline must be installed.

$$\begin{cases} z_h \leq z \\ z_h \leq 1 - u_h \end{cases} \tag{15}$$

There is also the possibility of installing new purification units. In this case, it is enough that there is any flow entering or leaving this unit. In this case, a binary variable $z_{kn}$ is associated with the installation of a new purifying unit and the logical constraints can be expressed by:

$$\begin{cases} FK_k \geq \varepsilon \times z_{kn} \\ FK_k \leq \left(FPur_{max,k}\right) \times z_{kn} \end{cases} \quad \forall k \in NHP \tag{16}$$

The same procedure for installing new compressors was also done (constraints in Equations (12) and (13)) if it is necessary to install new compressors on streams involving a new PSA.

### 3.2.5. Operating Costs

Operating costs include the production of hydrogen, the cost of electricity used in compressors, the operating cost of the purifying units, and the economic value corresponding to the burning gas in the fuel system. The cost of hydrogen production is assumed directly proportional to the flowrate, and it is defined as follows:

$$CH2I = \sum_{i \in HS} FH2I_i \times C_i \tag{17}$$

where $C_i$ is the cost of producing hydrogen. The electricity cost of the compressor is directly proportional to the power ($W$):

$$W = F \times w \tag{18}$$

where $W$ is the power of the compressor with the flowrate being compressed $F$, $w$ is the intensive power estimated from the stream properties ($C_P$, $C_V$, z), the inlet and outlet pressure, and the compressor efficiency [11].

$$w = \overline{(C_P \times T / \eta)} \times \left[ \left( \frac{P_{out}}{P_{in}} \right)^{\frac{\gamma - 1}{\gamma}} - 1 \right] \times (\rho_o / \rho) \tag{19}$$

where $C_P$ is the heat capacity, $T$ is the stream temperature, $\eta$ the efficiency of the compressor, $P_{out}$ and $P_{in}$ are the outlet and inlet pressure, respectively, $\rho_o$ and $\rho$ are the densities at design conditions and standard conditions, respectively, $\gamma$ is the ratio of the heat capacity at constant pressure to that at constant volume. For a given connection, e.g., $FIJ_{i,j}$, the corresponding intensive power $w_{i,j}$ is previously calculated as a model parameter. For the complete model, the total electricity cost is calculated by the following Equation (20). The indices $\alpha$ and $\beta$ represents the possible connections involved (*i,j; j,k; k,j; j,j'; i,k; i-waste; j-waste; k-waste*):

$$CH2C = \left( \sum_{\alpha} \sum_{\beta} F_{\alpha,\beta} \times u_{deltaP\alpha,\beta} \times w_{\alpha,\beta} \right) \times C_{eletric} \tag{20}$$

where $C_{eletric}$ is the electricity cost. It is worth to note that each term is multiplied by the binary parameter $u_{DeltaP}$ (1 if the pressure ratio is higher than one), for the cases in which the flowrate is not zero, but there is no need for compression. It does not matter if a new compressor is installed or

an existing compressor is used, both consumes energy. Equation (20) will compute the energy cost correctly, and it takes into account the electricity used in existing and new compressors.

The cost of purifying unit is proportional to the feed flowrate:

$$CH2K = \sum_{k \in HP} FK_k \times C_k \tag{21}$$

where $C_k$ is the cost of using the PSA purification units, new and existing ones.

The economy value corresponding to the burning of excess purge flows is corresponding to the cost of hydrogen and methane used as fuel and calculated as:

$$CH2F = C_{fuel} \times F \times (y \times \Delta H°_{H2} + (1 - y) \times \Delta H°_{CH4}) \tag{22}$$

where $C_{fuel}$ the cost per unit of energy, $F$ is the gas flowrate, and $y$ is the hydrogen composition. Assuming a binary mixture, $1 - y$ represents the methane composition. The parameters $\Delta H°_{H2}$ and $\Delta H°_{CH4}$ are the standard heat of combustion of hydrogen and methane, respectively. For the complete model, taking into account the total contributions, the economic value corresponding to the total cost of fuel is calculated as follows:

$$CH2F^T = C_{fuel} \times \sum_{\alpha} F_\alpha \times [y_\alpha \times \Delta H°_{H2} + (1 - y_\alpha) \times \Delta H°_{CH4}] \tag{23}$$

The subscript $\alpha$ denotes all units sending streams to the fuel system $(i, j, k)$. Since it corresponds to a saving cost, this value must be subtracted from the total operating cost. The operating cost parameters assumed in this work are presented in Table 1. The assumed values were the same used in example 1 [11], a case study of this work, also chosen based on the reviewed articles.

**Table 1.** Parameters used to calculate the operating cost.

| | | |
|---|---|---|
| Hydrogen cost—$H_2$ plant | $C_i$ | 0.07 \$/Nm$^3$ |
| Hydrogen cost—CCR | $C_i$ | 0.08 \$/Nm$^3$ |
| Electricity cost | $C_{eletric}$ | 0.03 \$/kWh |
| Purification cost | $C_k$ | 0.0011 \$/Nm$^3$ |
| Fuel cost | $C_{fuel}$ | 2.5 \$/MMBtu |

### 3.2.6. Investment Costs

The capital cost includes the cost of new compressors ($C_{new\ compressor}$), new purification units ($C_{new\ PSA}$) and new pipelines ($C_{piping}$). Hallale and Liu [11] describe the cost for the inclusion of new compressors for a particular flowrate, with a fixed cost with a binary variable and a variable cost associated with the flow:

$$C_{new\ compressor} = a \times z_c + b \times W \tag{24}$$

$W$ is calculated by Equation (18) and $z_c$ is the binary variable associated with the installation of a compressor for the corresponding flow and multiplied the fixed part of the new compressor cost, so it is considered only when the compressor is installed. The complete equation for accounting the new compressor cost is given by Equation (25). The indices $\alpha$ and $\beta$ represents the possible connections involved $(i,j; j,k; k,j; j,j'; i,k; i\text{-}waste; j\text{-}waste; k\text{-}waste)$.

$$\begin{aligned}
C^T_{new\ compressor} &= a \times \left( \sum_\alpha \sum_\beta z_{c\alpha,\beta} \right) \\
&+ b \times \left( \sum_\alpha \sum_\beta F_{\alpha,\beta} \times u_{deltaP\alpha,\beta} \times w_{\alpha,\beta} \times \left(1 - u_{c\alpha,\beta}\right) \right) \times C_{eletric}
\end{aligned} \tag{25}$$

The cost associated with the installation of new piping is described below, including a fixed part with a binary variable and a variable part dependent on flowrate. For these calculations, it is necessary to inform the distances between the already installed units of design.

$$C_{new\ piping} = \left(c \times z_h + d \times D^2\right) \times L \tag{26}$$

With

$$D^2 = (4 \times F/\pi \times \vartheta) \times (\rho_o/\rho) = (4 \times F/\pi \times \vartheta) \times \left(\frac{T}{T_0}\right) \times \left(\frac{P_0}{P}\right) \tag{27}$$

where $L$ is the pipe length [m], $c$ and $d$ are constants, $\vartheta$ is the gas surface velocity (usually 15–30 m/s; assumed an average value of 22.5 m/s in this work), and $D^2$ is the equivalent square diameter [11]. The binary variable $z_h$ indicates the need to install the new pipeline. Equation (27) is replaced in Equation (26) in order to express the cost of piping as a function of the flowrate. The equation for the model (total cost of new piping) is represented by Equation (28). The indices $\alpha$ and $\beta$ represents the possible connections involved (*i,j; j,k; k,j; j,j'; i,k; i-waste; j-waste; k-waste*). Each term is multiplied by $\left(1 - u_{h\alpha,\beta}\right)$ in order to consider only the cost of new piping.

$$\begin{aligned} C_{new\ piping}{}^T = c \times & \left( \sum_\alpha \sum_\beta z_{h\alpha,\beta} \times L_{\alpha,\beta} \right) \\ & + d \times \left( \sum_\alpha \sum_\beta F_{\alpha,\beta} \times L_{\alpha,\beta} \times w \times \left(1 - u_{h\alpha,\beta}\right) \times \left(\frac{T}{T_0}\right) \times \left(\frac{P_0}{P}\right) \right) \end{aligned} \tag{28}$$

There is also the possibility of installing new purification units. For this case, the cost of a PSA unit (purifier considered in this work) is a linear function of the unit flowrate (variable part) and include binary variable corresponding to the fixed installation cost:

$$C_{new\ PSA} = a_{PSA} z_{kn} + b_{PSA} \times F_{in,\ PSA} \tag{29}$$

where $a_{PSA}$ and $b_{PSA}$ are constants and $F_{in,\ PSA}$ is the inlet flowrate of the PSA unit (MMscfd). The binary variable $z_{kn}$ is associated with the installation of a new purifying unit. The model equation is described as:

$$C_{new\ PSA}{}^T = a_{PSA} \sum_{k \in NHP} z_{kn} + b_{PSA} \times \left( \sum_{k \in NHP} FK_k \right) \tag{30}$$

This cost is only considered for new purifying units. The capital cost parameters used in this work are presented in Table 2. Different coefficients exist for calculating capital costs, including variations in temperature and materials involved. The most frequently used data in the reviewed papers were used, following Hallale and Liu [11]. The objective is to facilitate the comparison of the results obtained.

**Table 2.** Parameters used to calculate the capital cost [11].

| | |
|---|---|
| Cost of new compressors (k$) | $115 + 1.91 \times W$ <br> $W$ in (kW) |
| Cost of new piping ($) | $(3.2 + 11.42 \times D^2) \times L$ $D^2$ (in$^2$) and $L$ (m) |
| Cost of new PSA (k$) | $503.8 + 347.4 \times F$ <br> $F$ in (MMscfd) |

*3.3. Formulation of the Optimization Problem*

Based on all the costs involved in managing the hydrogen network described in the previous section, annual operating and annual capital costs are defined as:

$$C_{operating} = (CH2I + CH2K + CH2C - CH2F) \times t \tag{31}$$

$$C_{capital} = \left(C_{new\ PSA} + C_{new\ piping} + C_{new\ compressor}\right) \times A_f \tag{32}$$

where $Af$ is the annualizing factor, and $t$ is the considered operating time of the plant in one year. The annualizing factor is defined by:

$$A_f = f_i \times (1 + f_i)^n / (1 + f_i)^n - 1 \tag{33}$$

where $n$ is the number of years of interest for the return on investment and $f_i$ is the interest rate. The Total Annual Cost (*TAC*) consists of the summation of the operating and investment cost:

$$TAC = C_{operating} + C_{capital} \tag{34}$$

For the retrofit case of existing networks, the economy saving used as economic criteria is calculated as:

$$E = C_{OP}^{actual} - C_{OP}^{new} \tag{35}$$

where $C_{OP}^{actual}$ and $C_{OP}^{new}$ are the operating cost of the actual and new networks, respectively. The payback time is defined by the ratio of the total investment cost and the economy saving, and the following equation can estimate it.

$$pt = \frac{C_{capital}/Af}{E} = \frac{\left(C_{new\ PSA} + C_{piping} + C_{new\ compressor}\right)}{C_{OP}^{actual} - C_{OP}^{new}} \tag{36}$$

The MILP model formulated in this work is described by the set of constraints (1, 2, 3–17, 20, 21, 23, 25, 28, and 30—HNS LM (Hydrogen Network Synthesis—Linear Model)). For process optimization, different objective functions can be chosen to be minimized. In this case, operating cost (31) for the retrofit case was chosen. The proposed model has the advantage of being a linear model, for which quite robust solvers can be used. However, the main drawback is that a compressor is associated with each possible connection individually in order to avoid nonlinear material balances to identify the composition of the stream being compressed. For this case, streams cannot be mixed to use the same compressor, and the resulting network may end up with more compressor units than an alternative nonlinear model, in which streams are allowed to mix.

### 3.4. Mathematical Model: MINLP Formulation

A nonlinear model was also developed. In this model, the compressors are considered as independent units that may be used to connect units that need compression (see Figure 1b). Different from the other units, the inlet and outlet pressure of each compressor are free variables. The maximum number of compressors to be considered is set in the superstructure modeling, and it is obtained in the model solution previously. In this model, streams are mixed to enter the compressor. Therefore, the hydrogen composition is unknown and must be treated as a variable. Besides, since no compressors are associated with each stream individually, the flowrates are only possible if the current origin pressure is higher than the destination pressure. For a particular flow $F$ with upper bound $F^{max}$, the constraints (37) ensure that flow is only possible for this case (higher pressure to lower pressure):

$$F \leq F^{max} \times (1 - u_{deltaP}) \tag{37}$$

Despite the possibility of generating networks with fewer compressors, the nonlinearity comes up with a more difficult problem to be solved that is very dependent on the initial guess, as will be discussed later.

In the MINLP model, the superstructure is a bit different from the one presented, as illustrated in Figure 1b. In this case, the compressor is considered a unit of the network and, therefore, can have the same source (the compressor outlet) and consumer (the compressor inlet) functionality and must be

present in the balance equations. The only nonlinearity in this model arises in the hydrogen balance in the inlet of the compressors because there is the merging of flows and, consequently, the product flow/purity. It is necessary to know the inlet composition because the outlet flow with this composition is sent to other units, and the hydrogen balances depend on this value.

The equations that describe the nonlinear model are described below. Equations (1), (3)–(9) of the linear model are replaced by the equations below, as compressors need to be considered in material balances. In sources, in addition to Equation (2), there is Equation (38), which describes the sum of flow rates from sources for consumers, purifiers, compressors ($FIC_{i,c}$) and for burning. Hydrogen from the source can be sent to all these units.

$$FH2I_i = \left( \sum_{j \in HC} FIJ_{i,j} + \sum_{k \in HP} FIK_{i,k} + FIW_i + \sum_{c \in HCP} FIC_{i,c} \right) \forall i \in HS \tag{38}$$

For consumers, global and component material balances are made, where $FCJ_{c,j}$ is the flowrate from the compressor to the consumers and $FJC_{j,c}$ is the flow rate from consumers to compressors. The sum of the flowrate at the entrance of each consumer corresponds to the sum of the flowrate from the source, the purifier, another different consumer, and the compressor.

$$FJ_j = \sum_{i \in HS} FIJ_{i,j} + \sum_{k \in HP} FKJ_{k,j} + \sum_{j \in HC} FJJ_{j,j'} + \sum_{c \in HCP} FCJ_{c,j} \ \forall j \in HC \tag{39}$$

The same is true for the hydrogen balance, where in addition to flowrates, purities are considered. Here there is the purity of the compressor ($YC_c$).

$$FJ_j \times YJ_j = \sum_{i \in HS} FIJ_{i,j} \times YI_i + \sum_{k \in HP} FKJ_{k,j} \times YK_k + \sum_{j \in HC} FJJ_{j,j'} \times YP_j \\ + \sum_{c \in HCP} FCJ_{c,j} \times YC_c \ \forall j \in HC \tag{40}$$

The sum of the outlet flowrate of each consumer corresponds to the sum of the flowrate that the consumer forwards to the burn (waste), to the purification unit, to another different consumer, and the compressor if necessary.

$$FP_j = FJW_j + \sum_{k \in HP} FJK_{j,k} + \sum_{j \in HC} FJJ_{j,j'} + \sum_{c \in HCP} FJC_{j,c} \ \forall j \in HC \tag{41}$$

The global material balance and for hydrogen is also applied for purifiers. The material balance corresponds to the sum of all flowrates at the entrance of the PSA, which include the flowrates from consumers, sources, and compressors. The purification unit, in turn, can send flow to consumers, compressors and can burn the excess (waste), which can be seen in Equation (42). Equation (43) corresponds to the hydrogen balance, considering the flows directed to the purifier and forwarded from the purifier. In addition to these equations, the purified flow rate must not exceed the PSA capacity (Equation (44)), and, through the recovery of the PSA, the flowrates that are sent for burning are obtained (Equation (45)).

$$\sum_{j \in HC} FJK_{j,k} + \sum_{i \in HS} FIK_{i,k} + \sum_{c \in HCP} FCK_{c,k} = \sum_{j \in HC} FKJ_{k,j} \\ + FKW_k + FKW_{rec,k} + \sum_{c \in HCP} FKC_{k,c} \ \forall k \in HP \tag{42}$$

$$\sum_{j \in HP} FJK_{j,k} \times YP_j \ + \sum_{c \in HCP} FCK_{c,k} \times YC_c + \sum_{i \in HS} FIK_{i,k} \times YI_i \\ = \sum_{j \in HP} FKJ_{k,j} \times YK_k + \sum_{c \in HCP} FKC_{k,c} \times YK_k + FKW_k \times YK_k \\ + FKW_{rec,k} \times YKW_k \ \forall k \in HP \tag{43}$$

$$\sum_{j \in HP} FJK_{j,k} + \sum_{i \in HS} FIK_{i,k} + \sum_{c \in HCP} FCK_{c,k} \le FPur_{max,k} \ \forall k \in HP \tag{44}$$

$$\left(\sum_{i\in HS} FIK_{i,k} \times YI_i + \sum_{j\in HP} FJK_{j,k} \times YP_j + \sum_{c\in HCP} FCK_{c,k} \times YC_c\right) \times (1 - rec_k)$$
$$= FKW_{rec} \times YKW_k) \ \forall k \in HP \tag{45}$$

where $FCK_{c,k}$ is the flow rate from compressors to purifier, $FKC_{k,c}$ is the flow rate from the purifiers to the compressors. Also, as the compressors are like units in the hydrogen network, material balances are made. The sum of the flow that enters the compressors is called $FC_c$, which consists of the sum of the flows from sources, consumers, and purifiers.

$$FC_c = \sum_{c \in HCP} FIC_{i,c} + \sum_{c\in HCP} FJC_{j,c} + \sum_{c\in HCP} FKC_{k,c} \ \forall c \in HCP \tag{46}$$

Therefore, any flow that enters the compressor must be directed to the consumers and purifications units. If necessary, some part of the compressor flow that is not used can be sent directly for burning.

$$\sum_{c\in HCP} FIC_{i,c} + \sum_{c\in HCP} FJC_{j,c} + \sum_{c\in HCP} FKC_{k,c}$$
$$= \sum_{c\in HCP} FCJ_{c,j} + \sum_{c\in HCP} FCK_{c,k} + FCW_c \ \forall c \in HCP \tag{47}$$

It is also necessary to carry out the hydrogen balance in the flows that make up $FC_c$.

$$FC_c \times YC_c = \sum_{c\in HCP} FIC_{i,c} \times YI_i + \sum_{c\in HCP} FJC_{j,c} \times YP_j \sum_{c\in HCP} +FKC_{k,c} \times YK_k \ \forall c \in HCP \tag{48}$$

In the same manner as in the MILP model, a binary variable z is associated with each possible flowrate, including the flowrates involving the compressor units, e.g., $FIC_{i,c}$, $FJC_{j,c}$, $FKC_{k,c}$, $FCJ_{c,j}$, $FKC_{k,c}$, and $FCW_c$. The corresponding constraints are as described by Equation (10). Also, binary variables are associated with new pipelines (Equations (13) and (14)) and for new PSA (Equation (15)). The binary variable $z_{c\alpha,\beta}$ are used to define if the compressor unit is installed assuming the value of 1, 0 otherwise. Differently from the MILP model, $zc$ is not defined over a pair of streams; it depends only on the compressor unit. $FC_c$ is associated with the flow of each compressor. Constraints (Equation (49)) is used to establish which compressors are used and their flow rates.

$$\begin{cases} FC_c \geq \varepsilon \times zc \\ FC_c \leq F^{max} \times zc \end{cases} \tag{49}$$

As the pressures vary in the nonlinear model, pressure restrictions must be included, which guarantees the compressor inlet and outlet pressures. They are formulated in the same format as the logical flow restrictions. For a given compressor unit, the inlet pressure is set as lower than the minimum pressure among the pressure of the mixed streams entering the compressor (Equation (50)). The outlet pressure is set as higher than the maximum pressure among the pressure of the streams, leaving the compressor according to the pressure of the stream destination (Equation (51)). It is important to mention that, due to the minimization of the energy cost associated with the compressor in the objective function, which is proportional to the pressure ratio ($PC_{out,c}/PC_{in,c}$), the inlet pressure is set as the minimum stream pressure entering the compressor $c$, and the outlet pressure as the maximum stream pressure leaving the compressor $c$.

$$\begin{cases} PC_{in,c} \leq PP_j + (P^{max} - PP_j) \times (1 - z_{j,c}) \\ PC_{in,c} \leq PI_i + (P^{max} - PI_i) \times (1 - z_{i,c}) \\ PC_{in,c} \leq PK_k + (P^{max} - PK_k) \times (1 - z_{k,c}) \end{cases} \tag{50}$$

$$\begin{cases} PC_{out,c} \geq PJ_j - P^{max} \times (1 - z_{c,j}) \\ PC_{out,c} \geq PK_k - P^{max} \times (1 - z_{k,j}) \\ PC_{out,c} \geq PW - P^{max} \times (1 - z_{c,w}) \end{cases} \tag{51}$$

where $PC_{in,c}$, and $PC_{out,c}$ are the compressor $c$ inlet and outlet pressures, respectively, the binary variable $z$ is associated with flowrates (i.e., $z_{i,c}$, $z_{j,c}$, $z_{k,c}$ …) and $P^{max}$ is the maximum pressure of the network used to make the constraints (50) and (51) redundant for the corresponding non-existent connection (the corresponding binary is set to zero due to the zero flowrate).

The operating and capital costs are calculated in the same way as in the linear problem, as well as the logical flow restrictions. The cost of hydrogen production is obtained by Equation (17), Equation (52) represents the electricity cost, Equation (53) represents the purification cost, and cost of fuel is represented in Equation (54).

$$CH2C = C_{eletric} \times \sum_{c \in HCP} FC_c \times (C_P * T/\eta) \times \left( \left( \frac{PC_{out,c}}{PC_{in,c}} \right)^{\frac{\gamma-1}{\gamma}} - 1 \right) \times (\rho_o / \rho) \tag{52}$$

$$CH2K = \sum_{k \in HP} (\sum_{j \in HC} FKJ_{k,j} + FKW_{rec} + FKW_k + \sum_{c \in HCP} FKC_{k,c}) \times C_k \tag{53}$$

$$CH2F^T = C_{fuel} \times \sum_{\alpha} F\alpha W_\alpha \times [y_\alpha \times \Delta H°_{H2} + (1 - y_\alpha) \times \Delta H°_{CH4}] \tag{54}$$

The subscript $\alpha$ denotes all units sending streams to the fuel system (*i, j, k, c*). Equation (55) represents the cost of new compressors and Equation (56) the cost of new piping.

$$
\begin{aligned}
C_{new\ compressor} = \quad & a \times z_{cnewc} \\
& + b \times FC_{cnewc} \times (C_P * T/\eta) \times \left( \left( \frac{P_c^{out}}{P_c^{in}} \right)^{\frac{\gamma-1}{\gamma}} - 1 \right) \times (\rho_o / \rho)
\end{aligned}
\tag{55}
$$

$$
\begin{aligned}
C_{new\ piping} = c \times \quad & \left( \sum_\alpha \sum_\beta z_{h\alpha,\beta} \times L_{\alpha,\beta} \right) \\
& + d \times \left( \sum_\alpha \sum_\beta F_{\alpha,\beta} \times L_{\alpha,\beta} \times w \times \left( 1 - u_{h\alpha,\beta} \right) \times \left( \frac{T}{T_0} \right) \times \left( \frac{P_0}{P} \right) \right)
\end{aligned}
\tag{56}
$$

The indices $\alpha$ and $\beta$ represents the possible connections involved (*i,j; j,k; k,j; j,j'; i,k; i-waste; j-waste; k-waste; i,c; j,c; k,c; c,j; c,k; c-waste*). The MINLP model formulated in this work is described by the set of constraints (1, 2, 11, 14, 15, 16, 17, 37–56). The objective function is described in Equation (31). This MINLP model will be named to facilitate the description of the results by HNS NLM (Hydrogen Network Synthesis—Nonlinear Model).

*3.5. Virtual Compressors*

The main difference between the MILP model and the MINLP model is how the compressors are treated. In MILP, the compressors are associated with each particular flowrate. In this case, the streams are not mixed. However, in the MINLP, the compressors are treated as independent units, not associated with a flowrate. Then the stream can be mixed to enter the compressor and split leaving the unit. Besides the class of the resulting model (either linear or nonlinear), the linear model may result in a network with more compressors and pipelines than the nonlinear model. Both the linear and the nonlinear formulation are capable of representing the hydrogen network, so what differentiates them is the issue of allowed linearity (which can be improved through this proposed technique), the linear model is simpler to solve, and the global optimum solution is guaranteed.

To overcome a large number of compressor units and further investment cost reduction, a strategy to reduce the use of this equipment was carried out through an algorithm based on non-real streams or virtual compressors, i.e., it is possible to rearrange the streams and compressors if the compressor capacities were not reached. This developed technique is one of the contributions of this work. Through it, the linear model becomes competitive, compared to the nonlinear model, due to its advantages.

There are two cases where it is possible to perform this unit reduction: (Option 1) when there are streams with different composition being compressed and forwarded to the same unit or (Option 2) when streams coming from the same unit are compressed and forwarded to different units, as can be seen in Figure 2. In other words, it is possible to group streams and use the same compressor, thus decreasing the fixed part of the new compressor capital cost, since the variable part is flow dependent and does not change. It is worth nothing that the fixed cost of piping is also minimized due to the rearrangement of the streams.

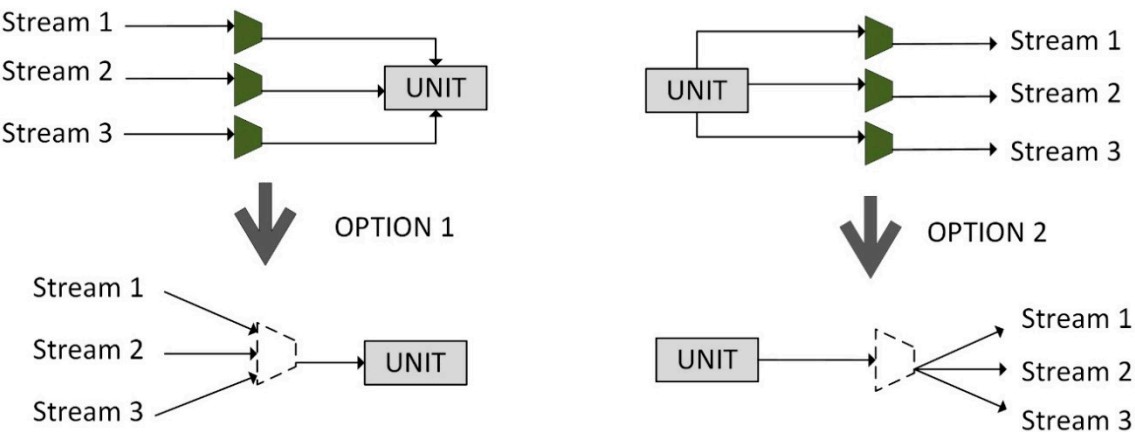

**Figure 2.** Virtual Compressor Approach–Possibilities of mixing streams in the compressors.

For each option, the inlet pressure (in Option 1) and the outlet pressure (in Option 2) must be corrected according to the minimum and maximum pressure of the involved streams, respectively. In that case, the energy cost and the variable part of the investment cost must also be recalculated. It should be noted that using this procedure, the solution is not unique, and the best solution is that with the maximum total cost reduction. Despite eventually unfavorable pressure changes, the number of compressor units can be reduced. Therefore, when this procedure is performed, the investment cost is almost always reduced. In this work, since the number of possible rearrangements is small, this procedure was performed by enumeration.

*3.6. Solution Strategy*

In this work, the MILP and the MINLP model were used to the network (re)design. Compared to the linear models, nonlinear models are wholly dependent on the initialization, which has a more challenging convergence. Also, for MILP models, the global solution can be obtained without a high computational effort. The MILP solution can be rearranged to reduce the compressor units, with the virtual compressors approach. Besides, the solution obtained by the MILP model can be used as a good and feasible initial point for the MINLP model. It is crucial to the grassroots designs since, in the retrofit case, the existing network can be used as an initial point. All these possibilities were evaluated in this work, and further discussion is presented in the results section.

The initialization strategy used can be described as follows:

1. The flowrates are fixed according to the existing network for the retrofit cases, and an LP subproblem with $F_{obj} = 0$ subject to the material balances is solved to obtain a feasible solution.
2. The binary variables ($z$) are initialized according to the existing network, i.e., $z = 1$, where there is a non-zero flowrate, $z = 0$ otherwise. Also, the other binary variables ($z_c$, $z_h$, $z_{kn}$) are fixed to zero, since they represent the installation of new compressors, piping, and purifying units.
3. The complete MILP model is solved. This result is defined as the existing network of each case study for later optimization (BASE CASE).
4. With all the variables values in the feasible solution defined by the existing network, the variables are set as free according to their lower and upper bounds. The complete MILP (HNS LM) is

solved (objective function = minimize operating cost). The MINLP (HNS NLM) proposed model is also solved to compare with the item (6).

5. The optimized network obtained through the linear model is evaluated with the rearrangement of compressors. Here the values of operating cost and capital differ between them due to the decrease in the number of compressors and the possible increase in electricity.

6. This network design is used to initialize the MINLP nonlinear model (HNS NLM).

For all cases, it was possible to ensure that the starting point was a feasible point. Figure 3 summarizes the initialization techniques performed.

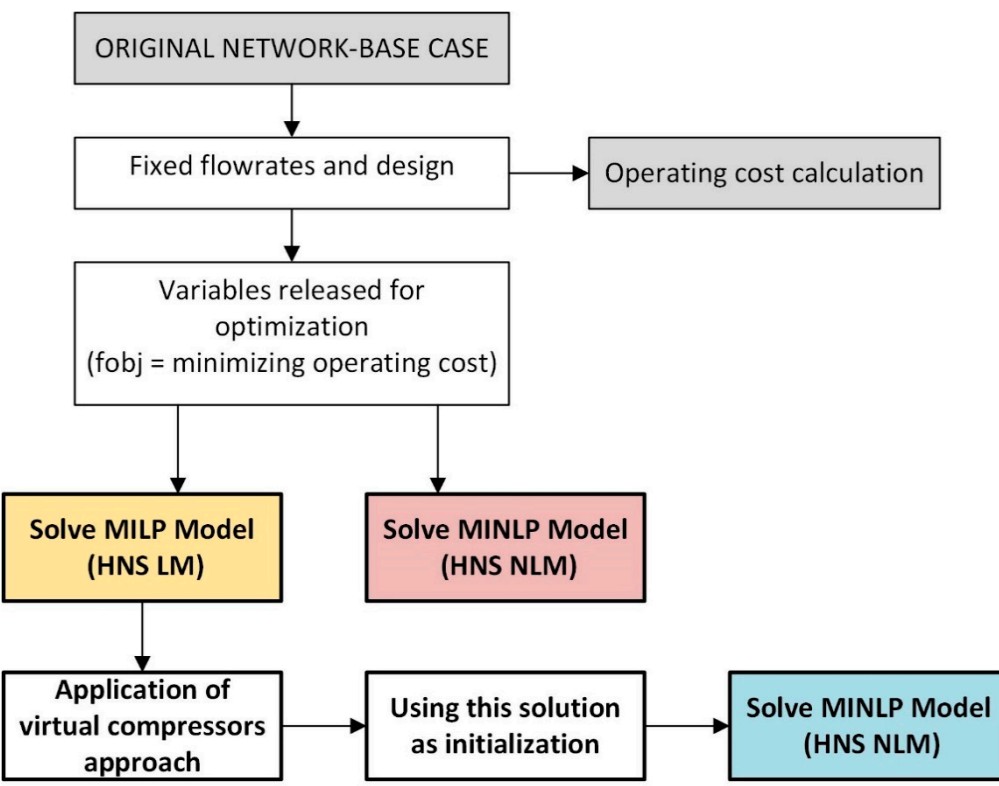

**Figure 3.** Summary of the methodology proposed in this article, through optimization via linear and nonlinear model.

## 4. Results

The model described in the previous section was validated using two examples of hydrogen networks proposed in the literature. The mathematical programming model was implemented in the modeling system GAMS 22.2 on a 3.6 GHz Intel® Core™ I7 CPU ( GAMS Development Corporation, Washington, DC, USA).The solver used to solve HNS LM was CPLEX (CPLEX 10, GAMS Development Corporation, Washington, DC, USA, 2006), and for HNS NLM it was DICOPT (/DICOPT 2x-C, GAMS Development Corporation, Washington, DC, USA, 2006).

For the case studies, it was considered the retrofit design for existing hydrogen networks. Therefore, the existing structure was explored considering the installation of new pipelines, new compressors, and purifying units. The economy saving is obtained by the operating cost reduction compared to the original solution. However, there is also an investment cost associated with non-existing equipment and pipelines. The payback time, i.e., the investment cost divided by annual operating cost savings was also used as an economic indicator for comparing the model solution.

The original network was ensured as a feasible starting point for all optimization problems. It was accomplished by fixing all the values of stream flowrates according to the existing network, and the total operating costs were calculated according to the parameters listed in this work for each case study.

For all cases, the original network was a feasible point. However, some authors have not presented the value of the parameters used to estimate the costs. Therefore, for a fair comparison, the costs were recalculated with the listed parameters in this work, and hence, despite the network configurations and flowrates are the same presented here, the costs are similar but not the same. Further discussion and considerations are given for each example.

### 4.1. Example 1

The first example is from Hallale and Liu [11]. The hydrogen network depicted in Figure 4 consists of a primary hydrogen production unit (H$_2$plant) and a secondary source, which is catalytic cracking (CCR). In this process, there are six consumer units: HC (hydrocracker), JHT (kerosene hydrotreater), CNHT (cracked naphtha hydrotreater), DHT (diesel hydrotreater), NHT (naphtha hydrotreater), and IS4 (hydrodealkylation). Two previously installed compressors are used, and there are no purification units. Flowrates are expressed in MMscfd (million ft$^3$/day, under standard conditions), stream purity, flowrates, and pressures are shown in Table 3.

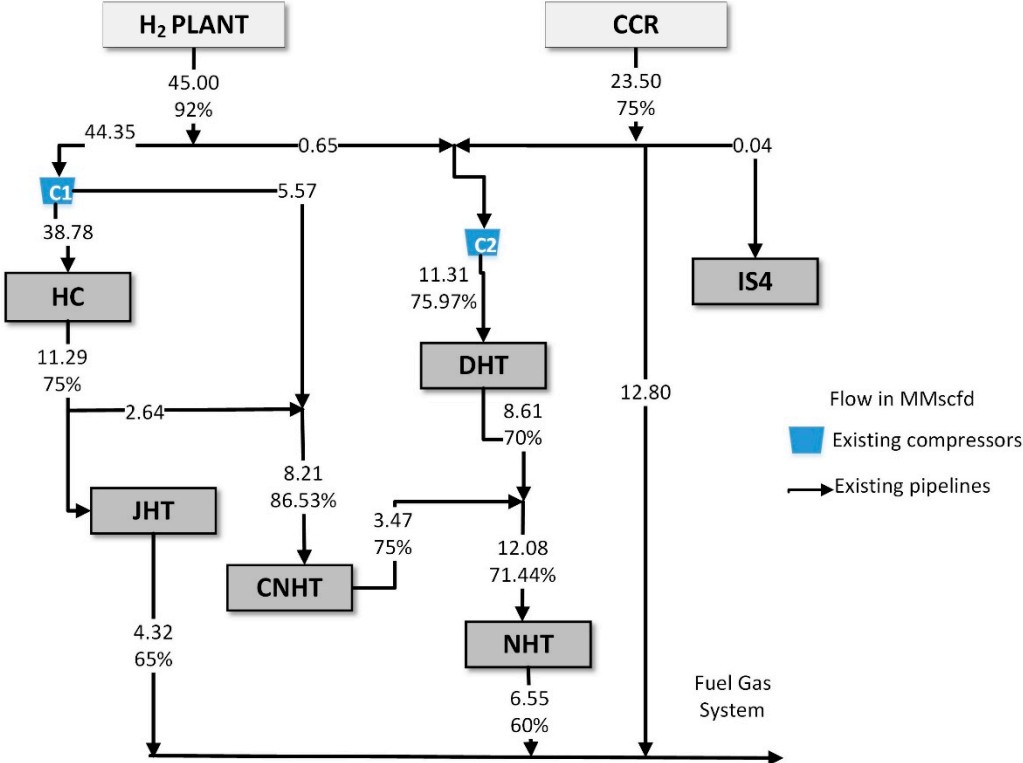

**Figure 4.** Existing hydrogen network for Example 1.

The objective function chosen for the problem analysis was to minimize the operating cost of the hydrogen network, Equation (31), using the parameters listed in Table 3 and the network configuration depicted in Figure 4. A variation of ±10% ($v_p$) in the nominal flow of consumers was allowed, $FJ_j$ and $FP_j$ were allowed in the original article. For the installation of a new PSA, the purity of 99.99% with a maximum operating capacity of 50 MMscfd, a recovery rate of 90%, and purge purity of 40.2% was considered.

**Table 3.** Flowrate, purity, and pressure information used in Example 1.

| Sources | $FH2I_i$ (MMscfd) | $FH2I_{i,max}$ (MMscfd) | $YI_i\%$ | $PI_i$ (psia) | | |
|---|---|---|---|---|---|---|
| H$_2$ plant | 45.00 | 50.00 | 92.50 | 300 | | |
| CCR | 23.50 | 23.50 | 75.00 | 300 | | |
| **Consumers** | $FJ_j$ (MMscfd) | $YJ_j\%$ | $PJ_j$ (psia) | $FP_j$ (MMscfd) | $YP_j\%$ | $PP_j$ (psia) |
| HC | 38.78 | 92.00 | 2000 | 11.29 | 75.00 | 1200 |
| JHT | 8.65 | 75.00 | 500 | 4.32 | 65.00 | 350 |
| CNHT | 8.21 | 86.53 | 500 | 3.47 | 75.00 | 350 |
| DHT | 11.31 | 75.97 | 600 | 8.61 | 70.00 | 400 |
| NHT | 12.08 | 71.44 | 300 | 6.55 | 60.00 | 200 |
| IS4 | 0.04 | 75.00 | 300 | | | |

The annual operating costs for the original network were estimated at 39.819 $/year. This solution is referred here as Hydrogen Network -BASE CASE (HN0). The Hydrogen Network -BASE CASE corresponds to the existing basic topology.

The HNS LM model has about 763 single equations, 323 single variables, and 227 discrete variables. Through linear optimization, savings of $11 million per year were achieved with a total investment of $16 million. In this case, 9 new compressors and 16 new pipelines were installed, as well as a new PSA (HN1). Nearly a 28% reduction in operating cost was achieved. This network is shown in Figure 5a. The HN1 optimized network MILP model only imports 26.5 MMscfd, and the original network uses 44.9 MMscfd of hydrogen from H$_2$ Plant, which represents a reduction of almost 41% in the amount of imported pure hydrogen.

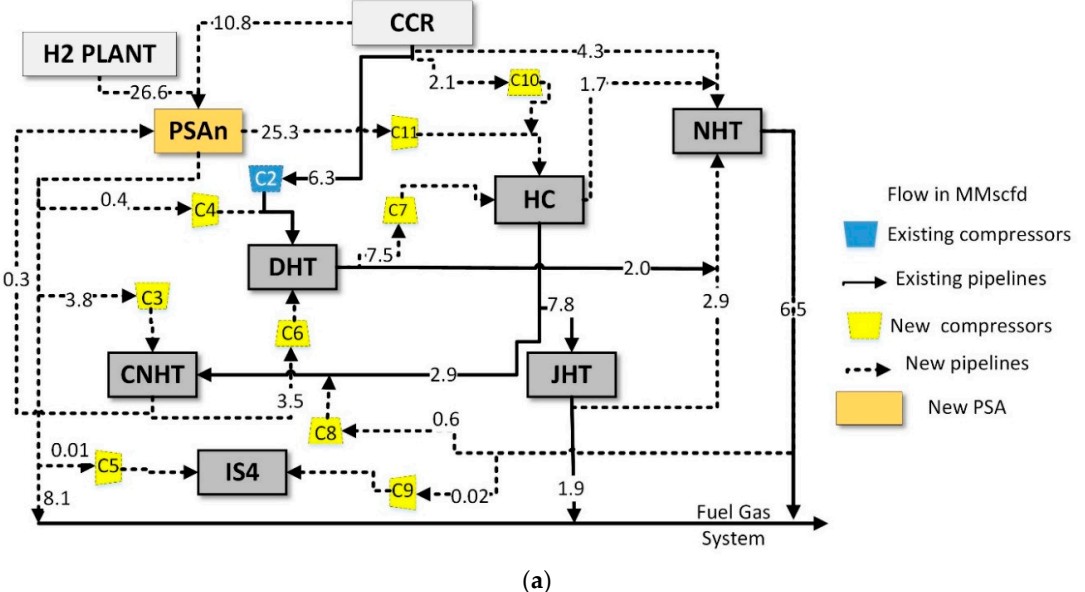

(**a**)

**Figure 5.** *Cont.*

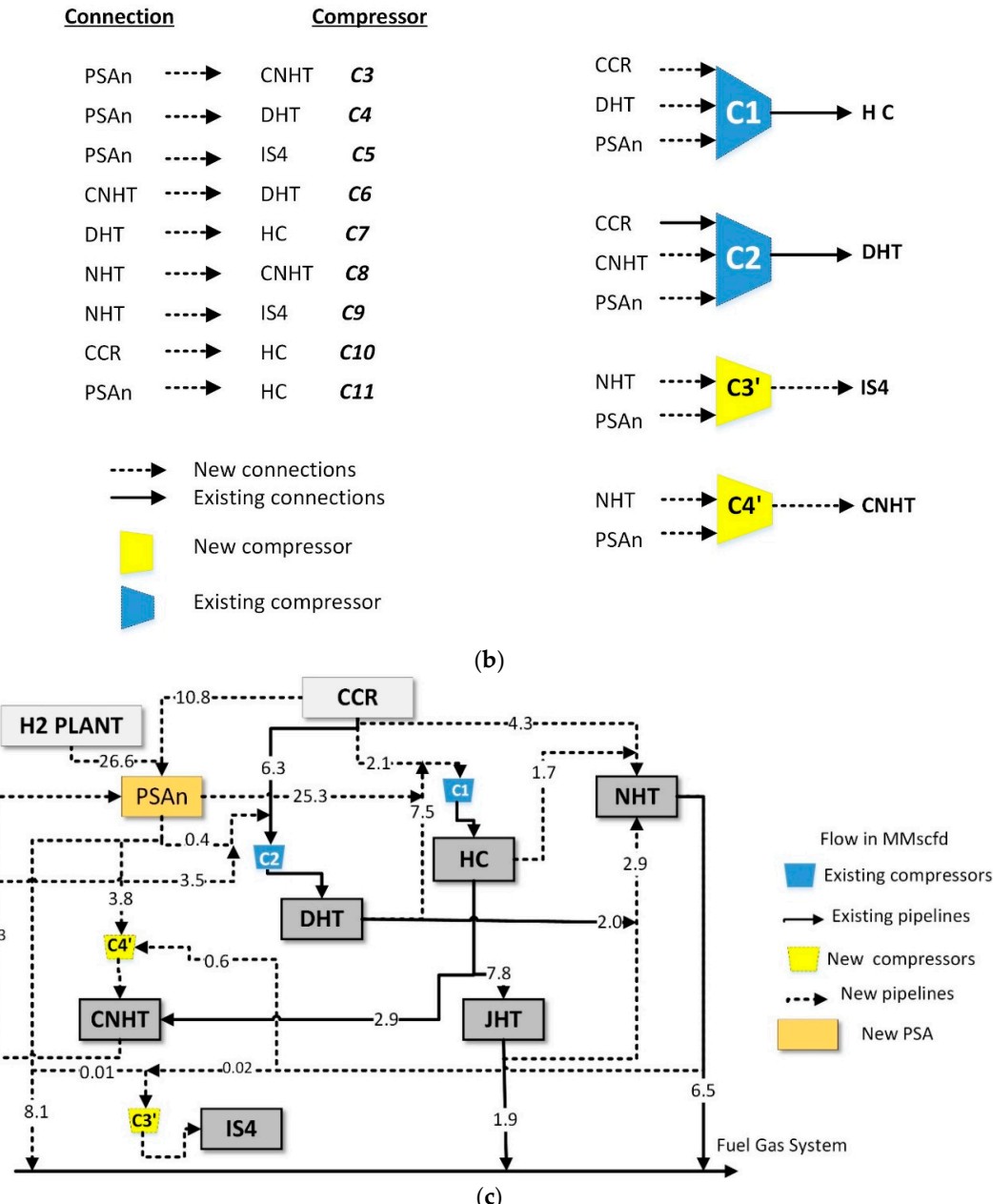

**Figure 5.** (**a**) Optimized network HN1 via HNS LM, for Example 1. (**b**) Virtual compressor approach applied to HN1 network. (**c**) Optimized network HN1′ with rearranged compressors.

In the HNS LM optimization, the merging of flows before the compressor units is not allowed. Therefore, the solution may result in a large number of installed compressors. However, the number of compressors can be reduced after the optimization, evaluating the obtained network, and, possibly, an even more significant cost reduction can be achieved. For the cases in which more than one stream leaving one unit is compressed and/or more than one stream is compressed to one unit, the streams can be rearranged to be compressed in a unique compressor unit saving the fixed cost associated to the compressor investment. According to the distance of the units, the cost of the pipeline must also be recalculated. As more than one alternative for the evolutionary network is possible, but they are only a few, this procedure can be executed manually by the designer. Therefore, we analyzed which

compressors were already previously installed based on the units and purity involved and if their nominal capacity allowed them to receive more streams. If positive, the stream was directed to it, and the new associated compressor could be eliminated. The rearrangement technique using virtual compressors applied to the compressors of example 1 can be seen in Figure 5b.

According to the optimization result (HN1), 9 new compressors were installed, which can be rearranged, as explained in Figure 2. According to option 1, where different flow rates that go to the same unit are grouped, rearranging in only 2 new compressors and using the two existing ones. The total cost of these new compressors is $0.271 million ($0.230 million of the fixed cost and $0.041 million of the variable cost), and this represents an 86.4% reduction in the total investment in new compressors. The total cost of piping also reduces by 40% due to the rearrangement of the compressors. This impact on total investment is 12.4% less.

It should be noted that as the compressors are rearranged, the inlet pressure is the lowest pressure between the flows. Therefore, the cost of electricity is slightly changed due to this, so the cost of electricity increased by 14.5% (from $0.136 million to $0.156 million) and an increase of 0.06% in operating costs. The proposed new topology can be analyzed in Figure 5c, and HN1 will represent that network.

To compare the linear and nonlinear formulations, the original network was optimized through the nonlinear model HNS NLM, described in Section 3.4. The first initialization used here was the original network, in example 1 (Figure 4). The HNS NLM model has 944 single equations, 473 single variables, and 308 discrete variables. The operating cost obtained was $28.183 million per year and $7.846 million per year of capital cost (one PSA and 10 pipelines), called network HN2. The result obtained in the two proposed optimized networks is very similar; however, the nonlinear has fewer connections (Figure 6a). The most significant portion of the cost of capital corresponds to the quantity to be purified. The optimization of HN2 network is an integer solution (not an optimal as in HNS LM), which usually happens in nonlinear problems as it is not possible to guarantee optimum global optimization.

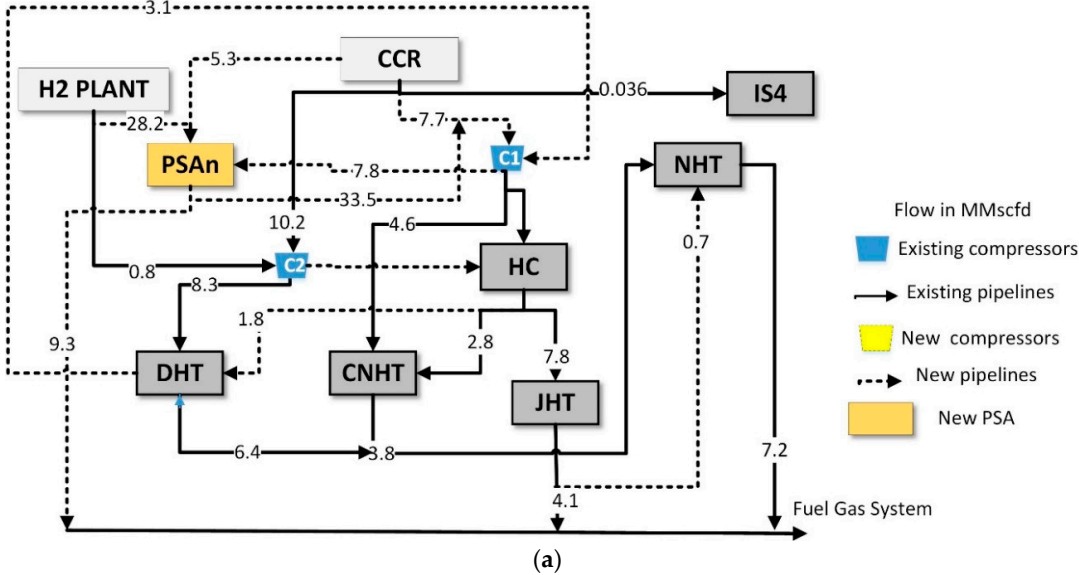

(a)

**Figure 6.** *Cont.*

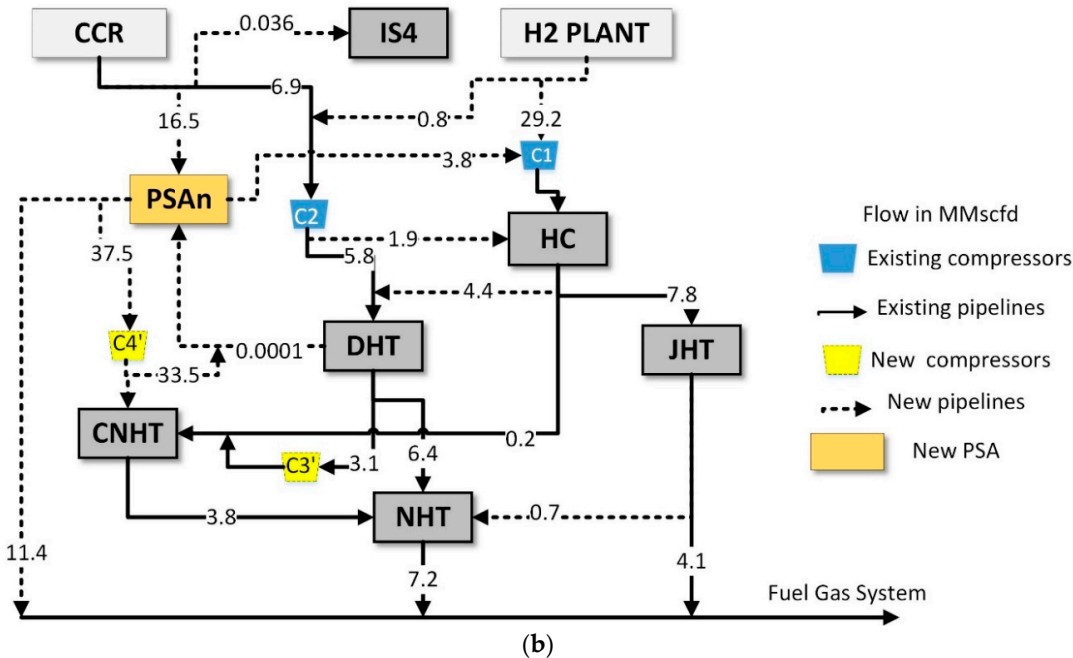

**Figure 6.** (**a**) Optimized network HN2 via HNS NLM for example 1. (**b**) Optimized network HN3 via HNS NLM with HNS LM as initialization, for example 1.

The second initialization made, which is the biggest contribution of this work, uses the result obtained from the HNS LM (HN1'- with compressors rearrangement) as the initialization of the nonlinear model HNS NLM, to facilitate the resolution of the nonlinear model. As already mentioned above, the HN1' network with the rearrangement of the compressors has a significant reduction in the cost of new compressors. For this reason, it is an excellent point option for the nonlinear model. Besides, as can be seen in the results, since nonlinear optimization has great locations, this initialization helped to improve the result. The HN3 network (obtained using MILP as a feasible point in MINLP) resulted in the lowest operating cost, a reduction of 31.2% (Figure 6b). However, comparing the payback, which refers to both the economy and the necessary investment, the network with the lowest payback is HN1'. This shows that with the HNS LM model, good and significant results are achieved, but through nonlinear optimization, less complicated networks with lower operational costs are achieved. For this, it is important to evaluate the design of the proposed network through different initializations.

All the results obtained in the different optimizations are summarized in Table 4. It is observed that the most significant reduction in the operation cost was obtained in the HN3 network. However, taking into account the investment and the payback time, the HN1' network proves to be an excellent alternative. Through the results obtained, it can be concluded that the two described models (linear and nonlinear) are efficient for the proposed optimization. The linear model is good enough and capable of providing considerably improved solutions. Besides, as an initial guess for the nonlinear model, it proved to be an even more competitive alternative. The compressor rearrangement technique provides a reduction in investments. When used to initiate the optimization of the nonlinear model, it provides designs with fewer lines and compressors.

**Table 4.** Results obtained in the different optimizations models for example 1.

| | COST ($\times 10^6$) | | | |
|---|---|---|---|---|
| | HNS LM | HNS LM | HNS NLM | HNS LM INITIALIZATION-HNS NLM |
| | HN1 | HN1′ | HN2 | HN3 |
| H$_2$ production ($/year) | 38.659 | 38.659 | 40.439 | 41.117 |
| Electricity ($/year) | 0.136 | 0.156 | 0.204 | 0.198 |
| Fuel ($/year) | 10.576 | 10.576 | 12.931 | 14.448 |
| Purification ($/year) | 0.429 | 0.429 | 0.470 | 0.568 |
| Operating cost ($/year) | 28.648 | 28.667 | 28.183 | 27.435 |
| New compressor ($/year) | 0.992 | 0.135 | - | 0.290 |
| New piping ($/year) | 0.415 | 0.405 | 0.419 | 0.341 |
| New PSA ($/year) | 6.801 | 6.801 | 7.426 | 8.937 |
| Capital cost ($/year) | 8.209 | 7.342 | 7.846 | 9.568 |
| Total capital cost ($) | 16.418 | 14.684 | 15.692 | 19.136 |
| TAC ($/year) | 36.857 | 36.009 | 36.029 | 37.003 |
| Economy ($/year) | 11.214 | 11.195 | 11.679 | 12.427 |
| Payback (year) | 1.464 | 1.312 | 1.344 | 1.540 |
| Resource time (s) | 0.040 | 0.040 | 1.337 | 5.427 |

As the original article of this case study does not present clear information about parameters and conditions used in the optimization [11], this work differs in values from the presented network. However, it is noteworthy that although the cost of the original network is different due to the explained, in this work, we considered the same calculation methodology for the original network (base case- HN0) and optimized network (HN1, HN2, HN3 … ), with specific parameters and conditions chosen.

The result obtained from the optimization in Hallale and Liu [11] is a 26.6% reduction in operating cost and payback time of 1.6 years, whose objective function was to reduce operating costs, limiting the payback time to 2 years. The achieved results obtained here with the proposed methodology are satisfactory as HN1 (HNS LM) optimized network reduced by 28.1% the cost of operating with a payback of 18 months. The optimized HN2 (HNS NLM) network achieved a 29.3% reduction in the operating cost with a payback time of 16 months, while Hallale and Liu [11], reduced operating cost by 15%, with a 17 months payback. For this reason, the result obtained was better than that presented in the original article, as in percentage, a more significant reduction in operating cost and payback was achieved. With the proposal to use the linear solution as a feasible point, HN3 network, the reduction was even higher (31% in operating cost), which shows the efficiency of the proposed technique.

*4.2. Example 2*

The second example used is from Sardashti Birjandi et al. [15]. The network is made up of two hydrogen producing units, a catalytic cracking plant (CCR) and a hydrogen generating unit (H$_2$plant), two purifying units (PSA), and 3 hydrogen consuming hydrotreating units (HDT I, HDT II, and HC),

as illustrated in Figure 7. In addition to the information, some parameters described in Table 5 are required. This HNS LM model has 524 single equations, 224 single variables, and 158 discrete variables.

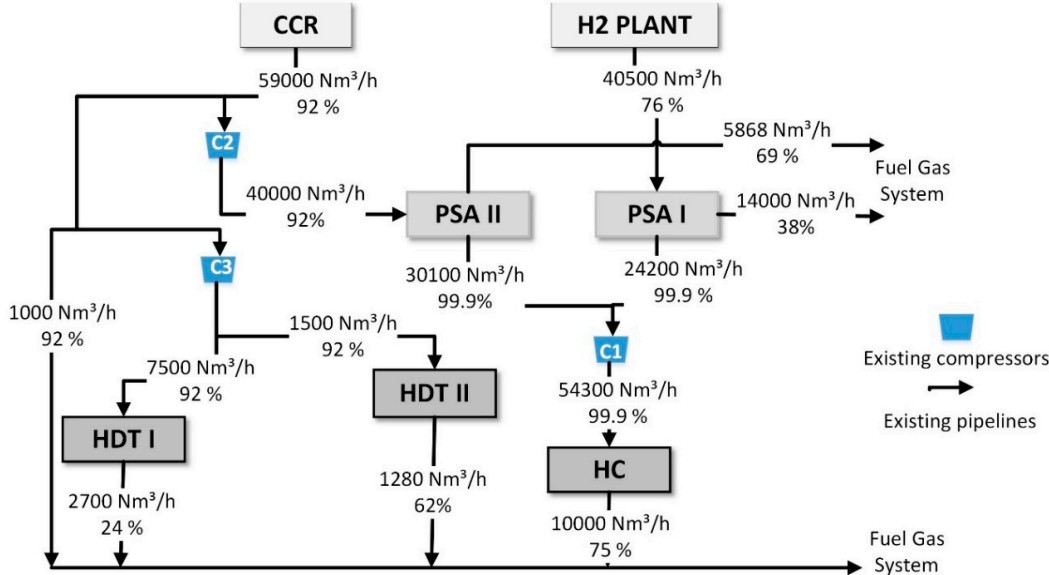

**Figure 7.** Existing hydrogen network for Example 2.

**Table 5.** Flowrate, purity and pressure information used in Example 2.

| Sources | $FH2I_i$ (Nm$^3$/h) | $FH2I_{i,max}$ (Nm$^3$/h) | $YI_i$ (%) | $PI_i$ (bar) | | |
|---|---|---|---|---|---|---|
| H$_2$ Plant | 40,500 | 90,000 | 76.00 | 22 | | |
| CCR | 59,000 | 65,000 | 92.00 | 4.50 | | |
| **PSA** | $FPur_{max,k}$ (Nm$^3$/h) | $YK_k$ | $YKW_k$ | Rec | | |
| PSA I | 80,000 | 99.90 | 38.00 | 0.85 | | |
| PSA II | 50,000 | 99.99 | 67.80 | | | |
| **Consumers** | $FJ_j$ (Nm$^3$/h) | $YJ_j\%$ | $PJ_j$ (bar) | $FP_j$ (Nm$^3$/h) | $YP_j\%$ | $PP_j$ (bar) |
| HC | 54,300 | 99.99 | 198 | 10,000 | 75.00 | 29.50 |
| DHT | 7500 | 92.00 | 55 | 2700 | 24.00 | 7.50 |
| NHT | 1500 | 92.00 | 55 | 1280 | 62.00 | 10.00 |

The annual operating costs for the original network were estimated at 44.017 $/year. This solution is referred here as Hydrogen Network -BASE CASE for example 2. This network corresponds to the existing basic topology (Figure 7).

Minimizing only the operating cost of the hydrogen network, savings around $12.4 million per year are achieved (HN4). For this design, the total investment of $22 million is paid off in 22 months. Six compressors, 10 new lines, and a new PSA were installed. The operating cost was reduced by 28.3%. To avoid the installation of a new PSA, the network has been further optimized (HN5), resulting in $11.7 million per year savings and with an even shorter payback time of approximately 2 months. Five new compressors and 6 new pipes were installed (HN5, Figure 8a). Almost a 26.5% reduction in operating cost was achieved.

In this case, when rearranging the compressors respecting the nominal capacity, there is a reduction from 5 new compressors to only 2 new ones and using the 3 already installed. The rearrangement technique using virtual compressors applied to the compressors of example 2 can be seen in Figure 8b. In terms of total compressor cost reduces from $1.194 million ($0.575 million fixed cost and $0.620

million variable costs) to $0.934 million ($0.230 million fixed cost and $0.704 million variable costs). It represents a 21.8% reduction in the total investment in new compressors. It is worth mentioning that the cost of electricity increased from $ 0.765 to 0.777 million per year due to the pressure drop in the rearrangement. The total investment cost reduces from $1.406 to $1.099 million. The proposed network design through the rearrangement of the compressors is represented by HN5′, as shown in the Figure 8c.

To make a more direct comparison with the retrofit results obtained in the original paper, the existing network was tested using the HNS NLM model described in Section 3.4. The HNS NLM has 787 single equations, 430 single variables, and 249 discrete variables.

The cost of operation in the nonlinear (HN6) problem is 4.7% lower than in the HNS LM problem (HN5). However, it is observed that the most significant difference is the amount sent to burning as fuel. The optimization of HN6 network is an integer solution, which usually happens in nonlinear problems as it is not possible to guarantee optimum global optimization. The design obtained in HN6 optimized network through an HNS NLM model is shown in Figure 9a. It is remarkable to highlight that the HN6 network has 3 new lines. However, the cost of piping in this problem is calculated as a percentage of the cost of capital, which in this case, is zero. Therefore, it is necessary to estimate an average value for the cost of piping, which can be obtained with the number of lines in previous examples. The average cost for 3 new lines is between $0.08 and $0.1 million per year, taking into account that the fixed part is the predominant value and does not vary much with the flow.

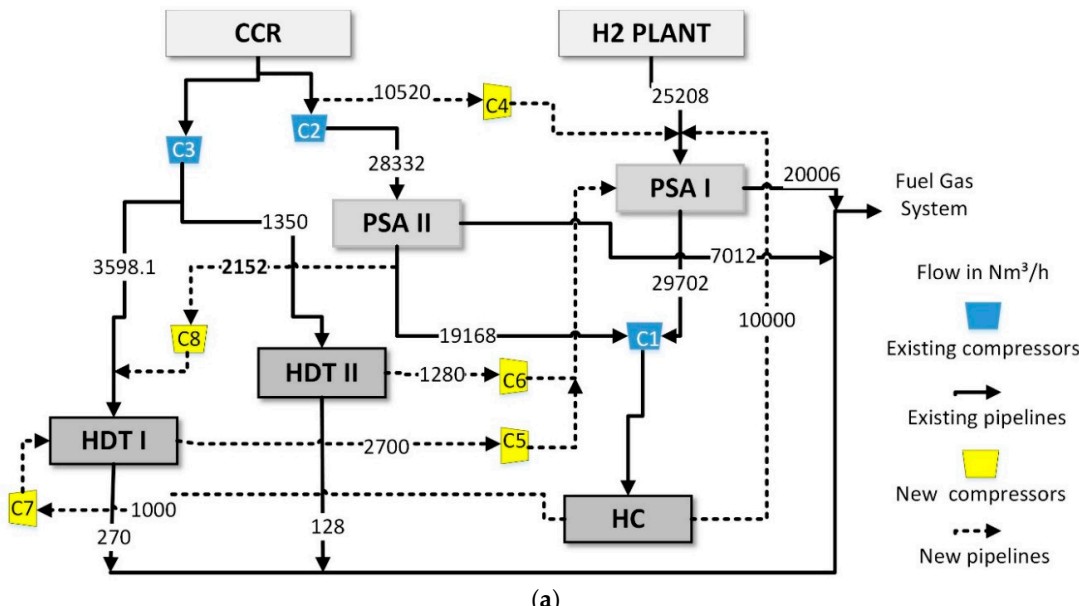

(a)

**Figure 8.** *Cont.*

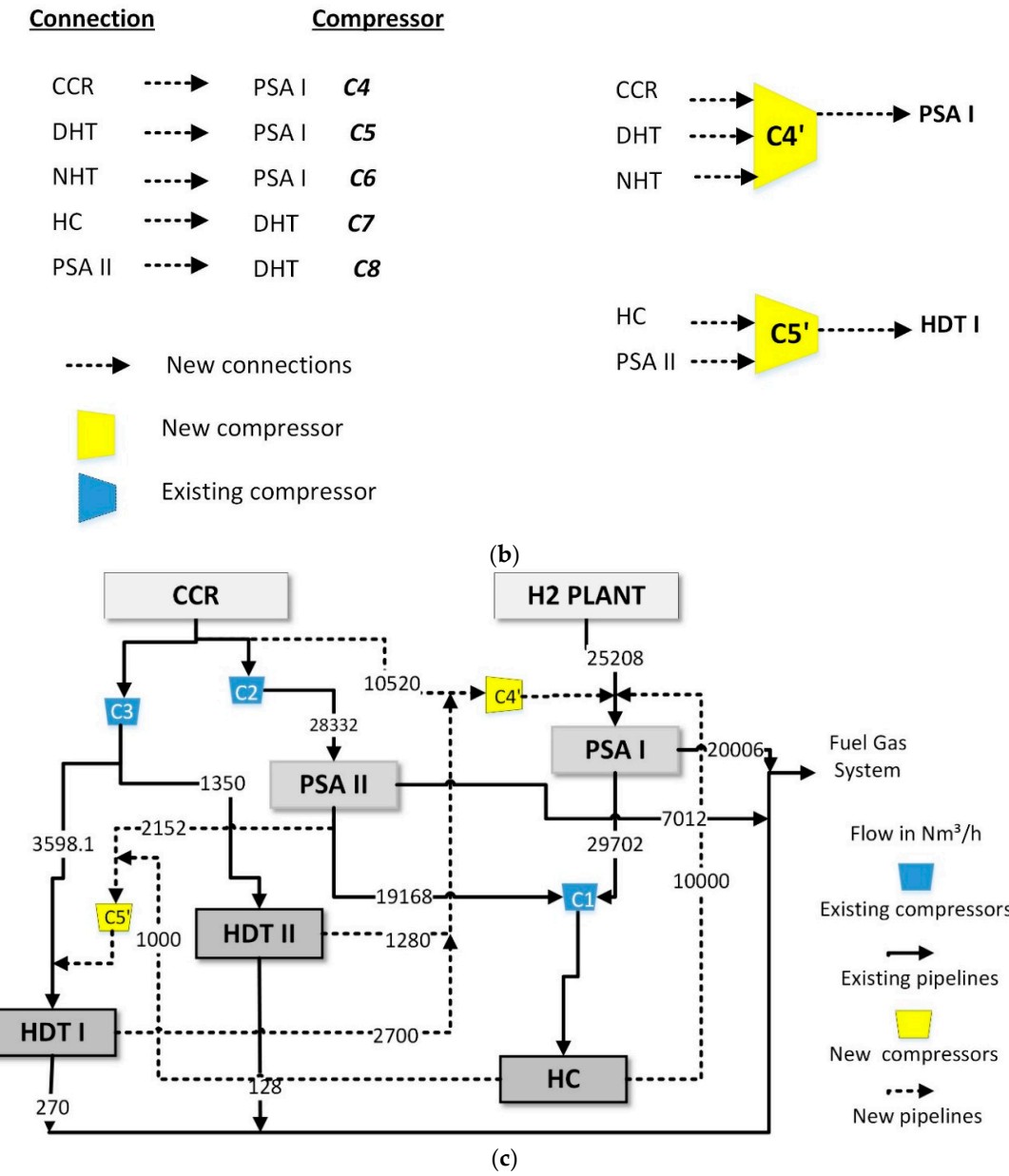

**Figure 8.** (**a**) Optimized network HN5 via HNS LM for Example 2. (**b**) Virtual compressors applied to HN5 network. (**c**) Optimized network HN5′ with rearranged compressors.

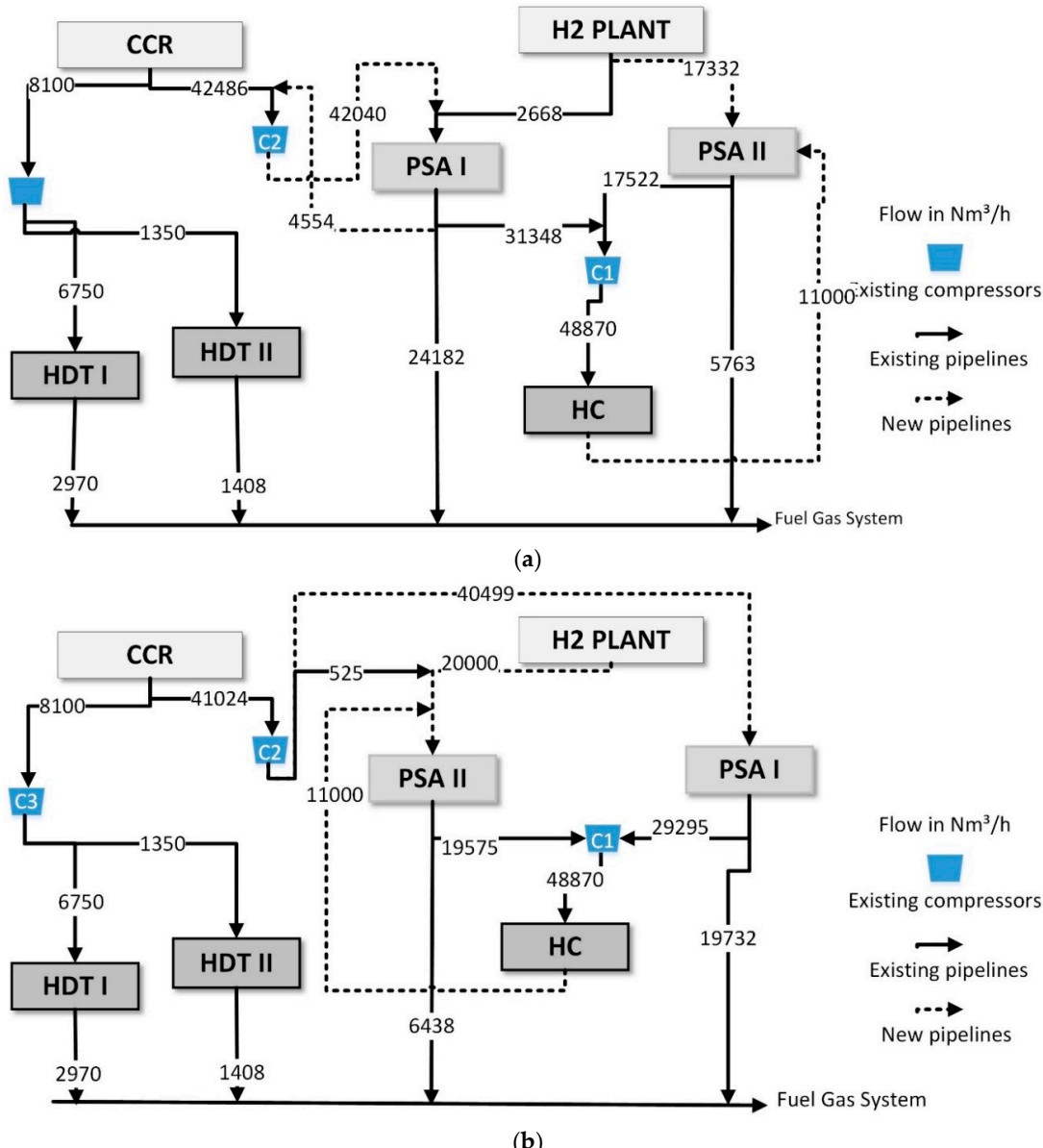

**Figure 9.** (**a**) Optimized network HN6 via HNS NLM for Example 2. (**b**) Optimized network HN7 via HNS NLM with HNS LM as initialization for example 2.

Using the same methodology as in example 1, the network optimized through the HNS LM (HN5′) was used as an initial value to solve the nonlinear problem. The idea of using the result obtained in the linear model to initialize the nonlinear model guarantees an even more significant reduction in operating cost, of 13.9%, with zero capital cost (despite 3 new lines). The initialization of the rearranged network generates better results, in addition to a network with fewer connections. The design network HN7 is shown in Figure 9b.

Table 6 summarizes the principal results obtained through linear and nonlinear models for example 2. The lowest operating cost is obtained with the initialization of the linear model in the HNS NLM resolution (HN7), in addition to presenting the advantage of easier convergence.

**Table 6.** Results obtained in the different optimizations models for example 2.

| | COST (×10⁶) | | | |
|---|---|---|---|---|
| | HNS LM | HNS LM | HNS NLM | HNS LM INITIALIZATION-HNS NLM |
| | HN5 | HN5′ | HN6 | HN7 |
| H$_2$ production ($/year) | 46.153 | 46.153 | 46.690 | 47.714 |
| Electricity ($/year) | 0.765 | 0.777 | 0.790 | 0.826 |
| Fuel ($/year) | 15.339 | 15.339 | 17.397 | 19.761 |
| Purification ($/year) | 0.762 | 0.752 | 0.723 | 0.803 |
| Operating cost ($/year) | 32.331 | 32.343 | 30.806 | 29.583 |
| New compressor ($/year) | 0.597 | 0.467 | - | - |
| New piping ($/year) | 0.105 | 0.082 | - | - |
| New PSA ($/year) | - | - | - | - |
| Capital cost ($/year) | 0.703 | 0.549 | - | - |
| Total capital cost ($) | 1.406 | 1.099 | - | - |
| TAC ($/year) | 33.034 | 32.892 | 30.806 | 29.583 |
| Economy ($/year) | 11.685 | 11.674 | 13.211 | 14.434 |
| Payback (year) | 0.120 | 0.094 | - | - |
| Resource time (s) | 0.067 | 0.067 | 4.495 | 1.906 |

In their original article, Sardashti Birjandi et al. [15] proposed the hydrogen network optimization through an MINLP model and obtained a 12% reduction in TAC. Considering TAC, this proposed HNS NLM model was able to reduce TAC by 30% (HN7), and the proposed HNS LM model was able to reduce TAC by 25.3%, which is a promising result. It is important to note that this case study was adapted from the example taken from the literature and that as many parameters are not described, the results would not be the same.

This example also shows that optimization through the linear model achieves considerable savings. Besides, as an initial guess for the nonlinear model, it proved to be an even more competitive alternative, further reducing operating costs.

## 5. Conclusions

In this work, an HNS LM (Mixed-Integer Linear Model) and HNS NLM (Mixed-Integer Nonlinear) optimization model is proposed for designing hydrogen networks for efficient use of this resource with cost reduction and environmental benefits.

The mathematical model is based upon superstructures, and it accounts for hydrogen sources, consumer units, purifying units, a fuel system, pressure constraints, and existing equipment and pipelines. The model can be used for grassroots designs and the retrofitting case. In the former, all the structure must be installed with an investment cost. In the later, the existing infrastructure is explored to reduce costs allowing the installation of new compressors, purifying units, and pipelines with an inherent investment cost. For both cases, the operating costs and the investment costs are the standard objective function to be minimized. Economic issues such as economy savings, maximum investment available, the payback time can be considered while delivering the optimal network design.

The model is thoroughly described, with all constraints, including the logical modeling equations used to accomplish design decisions and a proper estimation of costs, and all the model parameters. Initialization strategies for new design and retrofit cases were developed, which showed satisfactory results and efficiency for this work, both for existing and new networks.

The model was implemented in the modeling system GAMS solved with the solver CPLEX, and DICOPT and case studies from the literature were used to validate and explore the model features. For all examples, the proposed model was able to represent the existing networks as a feasible point, as well as to optimize them. Significant economic savings have been achieved when compared to existing networks, which shows that it is possible to work towards minimum hydrogen production and with investments payable in short periods.

The main breakthrough is the assumptions made in the mathematical modeling resulted in a linear model, which always converges to a global optimum, and it is speedy and robust. On the other hand, the drawback is that the solution may end up with a large number of compressor units. This issue can be overcome with the proposed algorithm basis evolution strategy to reduce the number of compressor units and pipelines and, therefore, the investment costs. This strategy has presented an excellent performance for the examples considered in this work. Besides, this technique can be extended to other problems of mass integration, such as pumps in water reuse, where the structure could also be represented through a linear model to facilitate resolution.

For comparison purposes, an HNS NLM model was also developed, in which streams can be mixed to be compressed at the same compressor unit. In this case, the number of compressors units is reduced when compared to the HNS LM model. However, the solution is influenced by the initial value, and it does not always converge, leading to a poor local minimum. The HNS NLM model also satisfies the needs of this work for the retrofit case and presented good results. However, the nonlinearity increases significantly the time need to solve the optimization problem. It is noteworthy that the HNS NLM model uses a superstructure that is different from the HNS LM, as the compressors are seen as a unit. The results obtained through nonlinear optimization compared to the linear ones, it has more flexibility of operation, because of the possibility of merging flowrates and share compressors. Resource time is not one of the main advantages when comparing linear with nonlinear. However, in the future, this work will be applied for multi-scenario optimization combined with production scheduling, so faster and more efficient resolution will be a critical issue.

For each case, different networks were proposed with different constraints. In general, the results were better than the original works of the case studies. Even though it was explored, the model versatility design networks allowing different constraints generating alternative designs according to the process requirements.

Different comparisons were made between the optimized networks in this work. With that, it can be concluded that the HNS LM model is satisfactory to optimize the hydrogen networks, even more with the rearrangement of the compressors, capable of reducing the investment costs. A reduction of 28% (example 1) and 26% (example 2) was obtained in the operating cost. In terms of the nonlinear model, the best results were obtained with the initiation of the network obtained from linear optimization. As a result, the operating cost was reduced by 31.2% (example 1) and 32.8% (example 2). This initialization technique was not found in the literature and proved to be an excellent tool for the optimization of hydrogen networks.

In this work, the importance of optimizing hydrogen networks is evident, aiming to minimize the operational cost. In addition, it is known that networks actually operate not only under nominal conditions as considered here, but also operate under different scenarios and different uncertainties. Since several factors affect this process, it is essential that the network must be able to work in various conditions. Therefore, the importance of working with uncertainties and multi-scenario optimization is evident. The MILP formulation proposed here can be easily extended to a multi-scenario version. In our future works, the uncertainty level will be addressed.

**Author Contributions:** Conceptualization, P.R.d.S.; Methodology, P.R.d.S., M.E.A., J.O.T. and L.F.T.; Software, P.R.d.S. and M.E.A.; Validation and Formal Analysis, P.R.d.S. and M.E.A.; Writing-Original Draft Preparation, P.R.d.S.; Writing-Review & Editing, P.R.d.S., M.E.A., J.O.T. and L.F.T.; Funding Acquisition, J.O.T. All authors have read and agreed to the published version of the manuscript.

**Funding:** This research was funded by ANP/Petrobras.

**Conflicts of Interest:** The authors declare no conflict of interest. The funders had no role in the design of the study; in the collection, analyses, or interpretation of data; in the writing of the manuscript, or in the decision to publish the results.

## List of Symbols

| | |
|---|---|
| $i, j, k, c$ | Sets of sources, consumers, purifiers, and compressors |
| $FH2I_i$ | Flowrate of hydrogen sources |
| $FH2I_{i, max}$, $FH2I_{i, min}$ | Maximum and minimum flow rate of hydrogen sources |
| $FIJ_{i,j}$ | Flowrate from source to consumer |
| $FIK_{i,k}$ | Flowrate from source to purifier |
| $FIW_i$ | Flowrate from source to waste (fuel system) |
| $FJ_j$ | Total consumer flowrate |
| $FKJ_{k,j}$ | Flowrate from purifier to consumer |
| $FJJ_{j,j'}$ | Flowrate from consumer $j$ to consumer $j'$ |
| $YJ_j$ | Consumer purity |
| $YI_i$ | Source purity |
| $YK_k$ | Purifier purity |
| $YP_j$ | Purge purity of consumer |
| $FP_j$ | Total purge consumer flowrate |
| $FJW_j$ | Flowrate from consumer to waste (fuel system) |
| $FJK_{j,k}$ | Flowrate from consumer to purifier |
| $FPur_{max,k}$ | Maximum capacity of the purifier |
| $FK_k$ | Total flowrate in the purifier |
| $FKW_k$ | Flowrate from purifier to waste (fuel system) |
| $FKW_{rec, k}$ | Purge flowrate from purifier to waste (fuel system) |
| $YKW_k$ | Purity of purge flowrate from the purifier |
| $F, F_{\alpha,\beta}$ | Flowrate |
| $F^{max}$ | Maximum flowrate |
| E | Parameter associated with the existence of flowrate |
| $z$ | Binary associated with flowrate |
| $z_c$ | Binary of a new compressor |
| $u_{deltaP}$ | Binary of the pressure difference between the units |
| $u_c$ | Parameter associated with existence compressor |
| $z_h$ | Binary variable from a new pipeline |
| $u_h$ | Parameter associated with existence pipeline |
| $z_{kn}$ | Binary variable from the new purifier |
| $\alpha, \beta$ | Represents the possible connections involved |
| $rec_k$ | Purifier recovery |
| $C_{operating}$ | Operating cost |
| $CH2I, C_i$ | Total and hydrogen production cost |
| $CH2K, C_k$ | Total and purification cost |
| $CH2C, C_{eletric}$ | Total and electricity cost |
| $W$ | Power compressor |
| $w$ | Intensive power compressor |
| $\overline{Cp}$ | Heat capacity |
| $T$ | Temperature |

| | |
|---|---|
| η | Compressor efficiency |
| $\gamma$ | Cp/Cv Ratio |
| $\rho_o$ | Density in standard condition |
| $\rho$ | Density |
| $P_{out}$ | Outlet pressure |
| $P_{in}$ | Inlet pressure |
| $CH2F^T$, $CH2F$, $C_{fuel}$ | Cost of burning purge as fuel |
| $y$ | Hydrogen fraction in the purge flow |
| $\Delta H°_{H2}$, $\Delta H°_{CH4}$ | Combustion heat of hydrogen and methane |
| $C_{new\ PSA}$, $C_{new\ PSA}{}^T$ | Cost of new purifier |
| $a_{PSA}$, $b_{PSA}$ | Parameters of new purifier cost |
| $C_{new\ piping}$, $C_{new\ piping}{}^T$ | Cost of new pipelines |
| $\vartheta$ | Superficial gas velocity |
| $L$ | Distance |
| $c$, $d$ | Parameters of piping cost |
| $C_{new\ compressor}$, $C_{new\ compressor}{}^T$ | Cost of a new compressor |
| $a$, $b$ | Parameters of new compressor cost |
| $t$ | Annual operating time |
| $Af$ | Annualized factor |
| $C_{capital}$ | Capital cost |
| $f_i$ | Interest rate |
| $TAC$ | Total annual cost |
| E | Economy |
| $C_{OP}^{actual}$, $C_{OP}^{new}$ | Actual and new operating cost |
| $pt$ | Payback |
| $FC_c$ | Total compressor flow |
| $FIC_{i,c}$ | Flow from source to compressor |
| $FCJ_{c,j}$ | Flow from the compressor to consumer |
| $YC_c$ | Purity in compressor |
| $FJC_{j,c}$ | Flow from consumer to compressor |
| $FCK_{c,k}$ | Flow from compressor to purifier |
| $FKC_{k,c}$ | Flow from purifier to compressor |
| $PC_{out,c}$ | Outlet pressure in the compressor |
| $PC_{in,c}$ | Inlet pressure in the compressor |
| $p^{min}$ | Minimum pressure |
| $p^{max}$ | Maximum pressure |
| $PI_i$ | Source pressure |
| $PK_k$ | Purifier pressure |
| $PW$ | Waste pressure |
| $PJ_j$ | Inlet consumers pressure |
| $PP_j$ | Outlet consumers pressure |

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
