# Peer review of "MILP Formulation for Solving and Initializing MINLP Problems Applied to Retrofit and Synthesis of Hydrogen Networks"

_processes, doi:10.3390/pr8091102_

Round 1

Reviewer 1 Report

GENERAL COMMENTS:

Authors consider an interesting and relevant problem (i.e, optimization issues in Retrofit and Synthesis of Hydrogen Networks)  in the field of energy management, and present an original study on the basis of a MILP and MINLP models to solve it. Authors consider the relevant literature, illustrate the proposed methodology and conduct a simulation study on some example cases.

The scientific work reported in the submitted manuscript is of clear interest under both the research and the application point of view. After a description of the specific energy management  issues and the related optimization needs, the proposed approach and a review of the relevant literature, authors present their modeling ideas illustrating their characteristics and potential benefits on the basis of some example cases. The manuscript presents a methodology which can be of potential interest also in different contexts, in particular where the integration of design solutions, demand, production and storage is relevant.

Overall, I believe that the paper presents an interesting study for a relevant application in the field of energy management. The topic of the paper seems to be sound with respect to MDPI-Processes journal, but some aspects of the manuscript need to be improved before publication can be considered.

In order to improve the overall quality of the paper, there are more detailed and specific suggestions and remarks reported in what follows.

Other comments and remarks:

Authors should discuss and motivate their deterministic approach. In fact, several aspects of the considered systems are in general affected by uncertainties and this point deserves to be addressed in the manuscript. In their introduction or (better) in the successive literature review and references, authors should also consider the role played by forecasting. For instance, I found the following recent work:

  • Bruno S., Dellino G., La Scala M., Meloni C., (2018) A Microforecasting Module for Energy Consumption in Smart Grids, 2018 IEEE International Conference on Environment and Electrical Engineering and 2018 IEEE Industrial and Commercial Power Systems Europe (EEEIC / I&CPS Europe), DOI: 1109/EEEIC.2018.8494345
  • The values of the parameters and coefficients used in the illustrative or computational examples need always some motivation. Readers are interested to know how these values are obtained, and possibly how the behavior of the methods is affected by these choices. So, these aspects deserve a better discussion in the manuscript.
  • As the manuscript discusses on a methodology which is of potential interest also in different contexts. Authors should better illustrate –at least in the conclusion– their ideas to extend/generalize their results on the basis of their characteristics, the potential strengths and weaknesses for further applications.
  • From the presentation point of view, even if the form is adequate, it is suggested that an additional proof reading be done by the authors before submitting a new version of the paper.
  • The first time authors use an initialism or acronym in their manuscript, the words should be written (please check) out with the short form placed in parentheses. This way, it's clear to the readers exactly what the letters mean.
  • The quality, clarity and readability of the figures should be improved. In addition, more explicative caption and descriptions should be used.

Concluding, on the basis of the aforementioned comments and remarks, the paper - in my opinion –  is of interest for publication on MDPI-Processes journal provided that authors improve their work considering the suggested revisions before the publication can be considered.

Reviewer 2 Report

This submission is about mixed-integer linear as well as nonlinear programming frameworks for hydrogen management during chemical refinery processes. Observations:
- Abstract minimally conveys the idea (while content and format issues are to be revisited).
--+ Introduction is basically insufficient. There are several references to offer a historical perspective. However, the presentation seems to be scattered and shallow, i.e., reorganize the section for relevant topical subcategories cohesively and present the corresponding references in each respective category. (One-reference paragraph can not serve for the purpose.)
- Clarify Line-151 (L-151) to state that "purely linear" problems converge to global optimums while problems such as MILP may not due to their nonconvex properties.
-- (3) is understandable in its logical terms. However, its component subscripts are very much inappropriate and confusing as there is a mismatch in (3) and the respective narratives, i.e., use plausible different subscripts for different consumers, etc., and rearrange (3) ((4) and (5)) accordingly. (In fact, revisit all equations to ensure matching/plausible index numbers.)
- Clarify L-288 for the 'unit cost of hydrogen" for the relevant flow rate.
- Properly and coherently integrate many short paragraphs.
-+ Based on previous results and a number of assumptions, MILP development/formulation is straightforward content-wise (not meant to overlook the efforts). Justifications for selections/assumptions under real-life operations are desired. MINLP development/formulation clarifies the nonlinearity concept within the considerations and presents the updated equations, mainly (38)-(48) (that should be justified/explained clearly and presented in a logical manner, e.g.,, for an equality, establish a logical separation and explain each term accordingly in the expression.
- Move (17) on L-473 into the body, etc.
-- Option-1 in the virtual compressor is easy to follow, provided that the real compressor capacity is not exceeded (When it is violated, which is likely, the corresponding detrimental terms need to be developed and incorporated in the model and optimization framework) . However, how to set different flow rates for Option-2? How to account for the time (potentially significant) to adjust the flow rates during the network stability and their transient effects for the hydrogen management efforts? "Mixing streams in the compressor" for MINLP. How important/significant or similar is this process in real life operations?
- Using MILP results as the initiation of MINLP framework sounds plausible (and they are). However, additional theoretical and numerical efforts should be done to convince its true potential (that goes well beyond intuitions). Based on the nonlinearity in the cost function and/or constraints, many awkward results could surface! The paragraph starting on L-657 is also concerning to provide a convincing rationale (other than intuitive or convenient one).
- Enhance Fig. 3 to incorporate the key initialization steps, etc. for more effective illustration.
- L-642-646 are concerning for a comparison of equals. If a genuine comparison is not possible, revise the narrative to focus on inherent (expected) benefits of the proposed approach.
-L-721-722 cost comment is concerning.
- Unexpected presentation issues: - Expand all acronyms in their first usage, - Ensure all-text abstract and conclusion, - Move the graphical abstract on the first page to a later page with a figure number, etc., - A missing period on L-50, - An unexpected new paragraph on L-53, - The model and programming reference on L-85 is confusing, - Grammatical issues on L-188, L-336, - A missing 'is' on L-234, - Remove/clarify vague references such as on L-255, - L-363 should refer to Table 3, - Table 2 seems to be missing, - Where is Af in (32)?, - WHat is "vi" on L-545?, - There seem to be two Table 3!,
